# The efficiency of transport into the stratosphere via the Asian and North American summer monsoon circulations

Xiaolu Yan[1], Paul Konopka[1], Felix Ploeger[1], Aurélien Podglajen[1], Jonathon S. Wright[2], Rolf Müller[1], and Martin Riese[1]

[1]Forschungszentrum Jülich (IEK-7: Stratosphere), Jülich, Germany
[2]Department of Earth System Science, Tsinghua University, Beijing, China

**Correspondence:** Xiaolu Yan (x.yan@fz-juelich.de)

**Abstract.** Transport of pollutants into the stratosphere via the Asian summer monsoon (ASM) or North American summer monsoon (NASM) may affect the atmospheric composition and climate both locally and globally. We identify and study the robust characteristics of transport from the ASM and NASM regions to the stratosphere using the Lagrangian chemistry transport model CLaMS driven by both the ERA-Interim and MERRA-2 reanalyses. In particular, we quantify the relative influences of the ASM and NASM on stratospheric composition and investigate the transport pathways and efficiencies of transport of air masses originating at different altitudes in these two monsoon regions to the stratosphere. We release artificial tracers in several vertical layers from the middle troposphere to the lower stratosphere in both ASM and NASM source regions during July and August 2010−2013 and track their evolution until the following summer. We find that more air mass is transported from the ASM and NASM regions to the tropical stratosphere, and even to the Southern Hemispheric stratosphere, when the tracers are released clearly below the tropopause (350−360 K) than when they are released close to the tropopause (370−380 K). For tracers released close to the tropopause (370−380 K), transport is primarily into the Northern Hemispheric lower stratosphere. Results for different vertical layers of air origin reveal two transport pathways from the upper troposphere over the ASM and NASM regions to the tropical pipe: (i) quasi-horizontal transport to the tropics below the tropopause followed by ascent to the stratosphere via tropical upwelling, and (ii) ascent into the stratosphere inside the ASM/NASM followed by quasi-horizontal transport to the tropical lower stratosphere and further to the tropical pipe. Overall, the tropical pathway (i) is faster than the monsoon pathway (ii), particularly in the ascending branch. The abundance of air in the tropical pipe that originates in the ASM upper troposphere (350−360 K) is comparable to the abundance of air ascending directly from the tropics to the tropical pipe ten months after (the following early summer) the release of the source tracers. The air mass contributions from the ASM to the tropical pipe are about three times larger than the corresponding contributions from the NASM. The transport efficiency into the tropical pipe, the air mass fraction inside this destination region normalized by the mass of the domain of origin, is greatest from the ASM region at 370−380 K. Although the contribution from the NASM to the stratosphere is less than that from either the ASM or the tropics, the transport efficiency from the NASM is comparable to that from the tropics.

# 1 Introduction

The structure and composition of the upper troposphere and lower stratosphere (UTLS) during boreal summer and fall over Asia and North America have many unique features. The Asian summer monsoon (ASM) and North American summer monsoon (NASM) anticyclones are not only important locally, but also affect first the entire northern hemisphere (NH) and then the whole atmosphere (e.g. Dethof et al., 1999; Randel et al., 2010, 2012; Yu et al., 2017). These circulations transport large amounts of water vapor and other chemical constituents from the surface to the UTLS (e.g. Rosenlof et al., 1997; Li et al., 2005; Park et al., 2009; Chirkov et al., 2016; Santee et al., 2017; Rolf et al., 2018). Air originating in the ASM and NASM is lifted high enough to ascend into the stratosphere and can be transported to distant locations (e.g. Vogel et al., 2016; Fadnavis et al., 2018), where it may exert considerable influence on UTLS chemistry (e.g. Cooper et al., 2006; Barth et al., 2012; Gottschaldt et al., 2018) and the radiative forcing of surface temperature (e.g. Solomon et al., 2010; Riese et al., 2012; Roy, 2018). Hence, transport of air uplifted by the ASM and NASM is an important factor influencing the composition of the UTLS and the evolution of global climate.

Transport via the ASM involves deep convection uplifting air from the boundary layer to the UTLS (e.g. Randel and Park, 2006; Wright et al., 2011; Bergman et al., 2012; Tissier and Legras, 2016), including the injection of boundary layer air by typhoons over Southeast Asia and Western Pacific into the outer regions of the ASM anticyclone (Vogel et al., 2014; Li et al., 2018). At higher levels, transport is dominated by the strong ASM anticyclone (e.g. Randel and Park, 2006; Park et al., 2008; Vogel et al., 2019), which is confined by easterly (on the tropical side) and westerly (on the extratropical side) jets. These jets act as strong transport barriers and restrict isentropic mixing (e.g. Ploeger et al., 2015; Poshyvailo et al., 2018). High concentrations of tropospheric tracers and low concentrations of stratospheric tracers within the anticyclone are evident in a variety of observations (e.g. Park et al., 2008; Glatthor et al., 2015; Ungermann et al., 2016; Santee et al., 2017; Höpfner et al., 2019). A fraction of the air within the ASM anticyclone eventually enters the tropical pipe and the NH extratropical lowermost stratosphere like a vertical 'chimney' or an isentropic 'blower' (Pan et al., 2016; Ploeger et al., 2017). The processes involved in this transport affect the chemical composition of the UTLS, including the buildup of ozone precursors (e.g. Fadnavis et al., 2015) and the formation of the Asian tropopause aerosol layer (ATAL) (Vernier et al., 2011; Yu et al., 2017; Vernier et al., 2018; Brunamonti et al., 2018). Recently, it has been suggested that a significant fraction of ATAL particles are composed of ammonium nitrate with ammonia from ground sources being the precursor pollutant (Höpfner et al., 2019). The ATAL has been estimated to cause a significant regional radiative cooling of the Earth's surface with an intensity of $\sim 0.1\,\mathrm{W\,m^{-2}}$ (Vernier et al., 2015).

In comparison to the ASM, transport to the stratosphere via the NASM has received relatively little scientific attention. Chen (1995) found that significant stratosphere-troposphere exchange (STE) occurs at potential temperatures greater than $340\,\mathrm{K}$ in the NH during boreal summer and linked this exchange to the ASM and NASM circulations. Dunkerton (1995) reported that the influence of monsoon circulations on STE vanishes above $\sim 25\,\mathrm{km}$. Water vapor retrievals from the Halogen Occultation Experiment (HALOE) satellite revealed two pronounced maxima over Asia and North America, presumably related to monsoon-driven convection over these two regions (Dethof et al., 1999; Randel et al., 2001). Dessler and Sherwood (2004)

pointed out that summer convection over North America is sufficient to exert a significant effect on the water vapor budget in the extratropical lower stratosphere, although not on the ozone budget. However, Randel et al. (2015) found that stratospheric water vapor in the ASM and NASM regions is mainly controlled by large-scale circulation and temperatures instead of over-shooting deep convection. Gettelman et al. (2004) argued that the NASM may entrain air from the high latitude troposphere into the subtropical lower stratosphere, but this air would rarely reach the deep tropics. Using satellite observations, Randel et al. (2012) linked a positive anomaly in the deuterium content of lower stratospheric water vapor to the NASM (and not the ASM) and showed that this anomaly extends into the tropics. The injection of water vapor into the stratosphere over the NASM may change the catalytic chlorine/bromine free-radical chemistry which could have implications for ozone loss (Anderson et al., 2012; Robrecht et al., 2019; Clapp and Anderson, 2019); however, Schwartz et al. (2013) found no indication of substantial ozone depletion triggered by injection of water vapor over the NASM region based on MLS data. Although water vapor in the stratosphere is strongly affected by the NASM, model simulations driven by the ERA-Interim reanalysis suggest that the fraction of NASM air in the NH midlatitude lower stratosphere is about one order of magnitude smaller than the fraction of ASM air (Ploeger et al., 2013).

Figure 1a shows higher tropopause altitudes over the ASM and NASM regions associated with anticyclonic circulations during July−August based on both the ERA-Interim and MERRA-2 reanalyses. Correspondingly, measurements of CO from the Aura Microwave Limb Sounder (MLS) show high values at $350\,\mathrm{K}$ over the ASM and NASM regions (Fig. 1b). Potential temperature−latitude sections of CO across the ASM ($90°\,\mathrm{E}$, Fig. 1c) and NASM ($90°\,\mathrm{W}$, Fig. 1d) anticyclones show that the enhancement of CO extends up to the UTLS with strong confinement over the ASM region and weak confinement over the NASM region. As described above, most previous studies have focused on the transport pathways related to the ASM anticyclone, with relatively few analyses of transport via the NASM anticyclone. In this study, we address the following questions:

(1) How does transport from the ASM and NASM regions affect stratospheric composition?

(2) How large are contributions from the ASM and NASM regions to the stratosphere relative to contributions from the inner tropics?

(3) From which levels within the ASM and NASM regions do the most effective transport pathways to the stratosphere originate?

Studies of air mass transport across the tropopause often rely on winds and diabatic heating rates from reanalyses. Diabatic heating rates, especially if derived from reanalyses, are more suitable to quantify vertical transport than are pressure tendencies, which are strongly affected by the numerical noise of the assimilation procedure (Eluszkiewicz et al., 2000; Schoeberl et al., 2003). However, diabatic heating rates are highly uncertain and differ substantially among reanalyses (Randel and Jensen, 2013; Wright and Fueglistaler, 2013; Abalos et al., 2015). These differences have important implications for transport calculations in the UTLS (Wright et al., 2011; Schoeberl et al., 2012; Bergman et al., 2013; Wang et al., 2014; Abalos et al., 2015; Ploeger et al., 2019; Tao et al., 2019). To evaluate the robust characteristics of transport in different reanalyses, we use both the ERA-Interim and MERRA-2 reanalyses to drive the Lagrangian transport model CLaMS. We then investigate the pathways for air mass transport from the ASM and NASM anticyclone into the lower stratosphere and quantify the associated transport

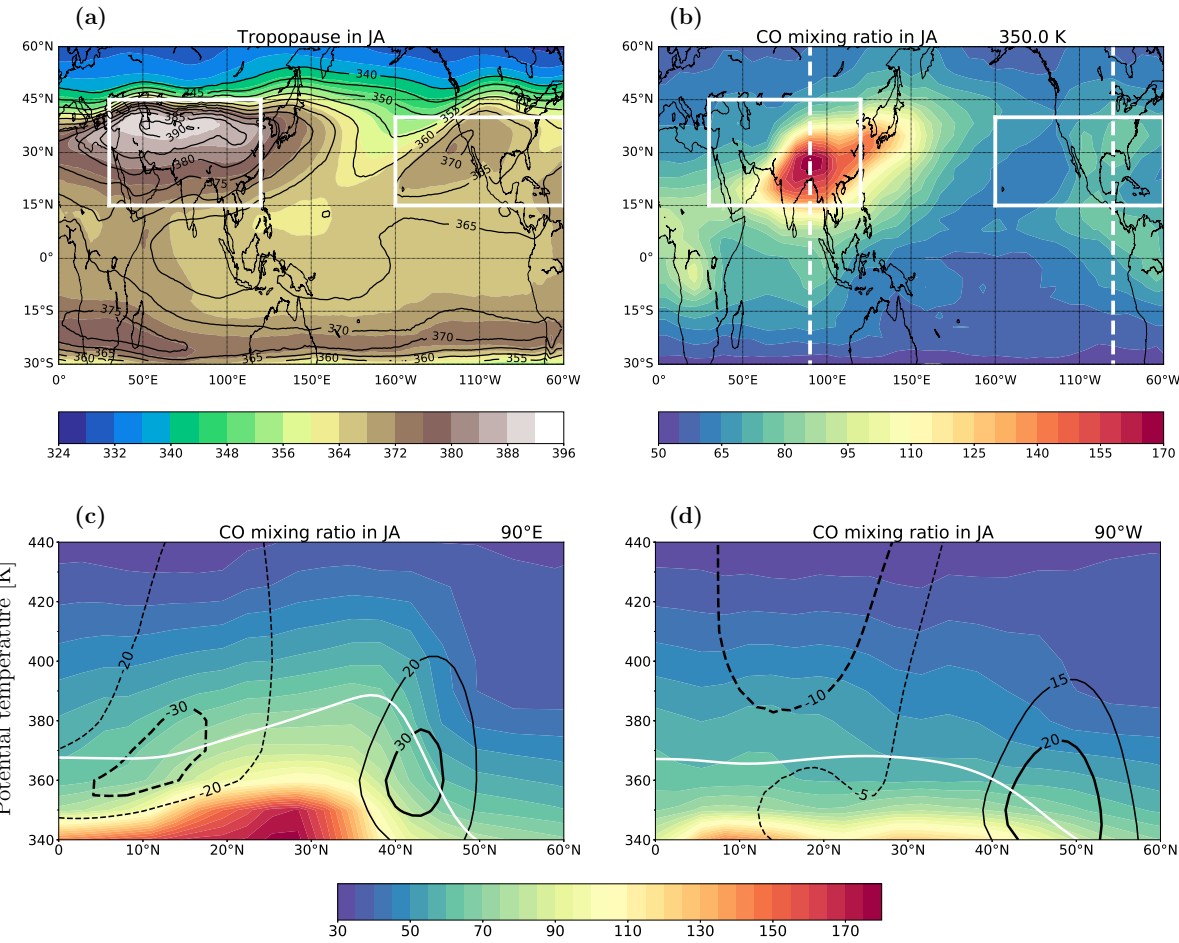

**Figure 1.** (a): Climatologies of the tropopause (lapse rate) potential temperature (in K) from ERA-Interim (1979−2016, color shading) and MERRA-2 (1980−2016, contours) for July and August. (b): Climatologies of CO (in ppbv) at $\theta$=350 K calculated from MLS (2004−2016) for July and August. (c,d): Potential temperature−latitude sections of CO (in ppbv, color shading) from MLS, tropopause height (white line) and wind (in m/s, black contours) from ERA-Interim for July and August during 2004−2016 along 90° E (c) and 90° W (d).

efficiencies and fractions of stratospheric air originating from each source region using both sets of simulations. In Sect. 3, we discuss reanalysis-related differences in the statistics of transport from monsoon regions to the extratropical lower stratosphere and the tropical pipe, and investigate the influences of the ASM and NASM on stratospheric composition (e.g. HCN, water vapor). In Sect. 4, we assess the efficiency of transport to the stratosphere via the ASM and NASM anticyclones. We discuss our findings in Sect. 5 before closing with a brief summary of the key results.

## 2  Data and methods

We apply tracer-independent diagnostics (i.e. without chemistry and emissions) to quantify air mass transport through the ASM anticyclone, NASM anticyclone and tropics. All diagnostics are based on simulations using the Lagrangian chemistry transport model CLaMS (McKenna et al., 2002; Konopka et al., 2004; Pommrich et al., 2014). The model transport is driven by horizontal winds and total diabatic heating rates from either the ERA-Interim (CLaMS-EI) or MERRA-2 (CLaMS-M2) reanalysis (for more details see Ploeger et al. 2019 or Tao et al. 2019). The use of two reanalysis data sets provides important constraints on the range of possible outcomes and potential biases associated with differences in horizontal winds and total diabatic heating rates between these two reanalyses.

We include artificial air mass origin fractions (AOFs; see also Orbe et al. 2013; Ploeger et al. 2017) to diagnose the fraction of air at any location in the stratosphere which left either the monsoon regions or the tropics during the previous boreal summer monsoon season. There are several ways to define the ASM anticyclone, including Montgomery stream function (Santee et al., 2017), stream function (Yan et al., 2018; Tweedy et al., 2018), geopotential height (Pan et al., 2016) and potential vorticity (PV; Ploeger et al., 2017). Applying similar criteria to the NASM is less straightforward.

Here, we follow the geographic definition of the ASM source region proposed by Yu et al. (2017), which covers the domain $[15° \text{N}, 45° \text{N}, 30° \text{E}, 120° \text{E}]$. Within this region, we consider several different layers, which collectively cover the atmospheric column from the middle to upper troposphere. For the layer spanning $370-380\,\text{K}$ potential temperature, our results are very similar to those of Ploeger et al. (2017), who used a PV-based approach to define the anticyclone (detailed comparisons are provided below). We therefore decide to use this simple geographical domain to represent the ASM region. One of the motivations for this approach is that it can be simply applied for the NASM region $[15° \text{N}, 40° \text{N}, 160° \text{W}, 60° \text{W}]$ and the tropics $[15° \text{S}, 15° \text{N}]$ as well. Vertically, we divide the column of air from the middle troposphere to the lower stratosphere above each region into 4 potential temperature layers $340-350\,\text{K}$, $350-360\,\text{K}$, $360-370\,\text{K}$ and $370-380\,\text{K}$ with $i$ labeling the layer.

$$M_{\text{ASM}}^{i}(\lambda, \phi, \theta, t) = \begin{cases} 1 & \lambda \in [30° \text{E}, 120° \text{E}], \phi \in [15° \text{N}, 45° \text{N}], \theta \in \text{Box}_i, \text{ and } t \in [\text{July, August}] \\ 0 & \text{elsewhere} \end{cases}$$

$$M_{\text{NASM}}^{i}(\lambda, \phi, \theta, t) = \begin{cases} 1 & \lambda \in [160° \text{W}, 60° \text{W}], \phi \in [15° \text{N}, 40° \text{N}], \theta \in \text{Box}_i, \text{ and } t \in [\text{July, August}] \\ 0 & \text{elsewhere} \end{cases}$$

$$M_{\text{Tropics}}^{i}(\lambda, \phi, \theta, t) = \begin{cases} 1 & \lambda \in [180°\text{ W, }180°\text{ E}], \phi \in [15°\text{ S, }15°\text{ N}], \theta \in \text{Box}_i, \text{ and t} \in [\text{July, August}] \\ 0 & \text{elsewhere} \end{cases}$$

Here $M_{\text{ASM}}^{i}$, $M_{\text{NASM}}^{i}$ and $M_{\text{Tropics}}^{i}$ mark source domains where the AOFs are set to 1 on each day during July−August in CLaMS simulations covering the period from 2010 to 2013 (the symbols $\lambda$, $\phi$ and t represent longitude, latitude and time, respectively). The AOFs are set to zero everywhere on 1 July of each year and then set to 1 inside the corresponding source

domains for this and every subsequent day through 31 August of the same year. The artificial tracer is advected as an inert tracer inside and outside of the source domains after its release, with advection driven by reanalysis horizontal winds and total diabatic heating rates. The AOF at any location in the stratosphere equals the mass fraction of air (in %) that left the corresponding source layer in the ASM, NASM or tropical domain during the previous monsoon season.

We run the simulation for the same period as Ploeger et al. (2017) to facilitate direct comparison between different definitions

of the domain boundary (simplified box and PV-barrier). July and August represent the mature phase of the monsoon. We simulate transport from July-August instead of June-September to exclude substantial in-mixing of air from areas adjacent to the monsoon regions during the transitional months when the monsoon circulation is spinning up or spinning down (June and September) and avoid overestimating transport (Sensitivity to the length of the initialization period is discussed in more detail later in the paper). After releasing the tracer in each box, we map the respective transport pathways from each source region

to the global stratosphere, focusing especially on transport to the tropical pipe (TrP), the extratropical lowermost stratosphere in the Southern Hemisphere (LS-SH), and the extratropical lowermost stratosphere in the Northern Hemisphere (LS-NH). Note that transport from each source region is simulated independently. Therefore, there are no interactive influences among transport from the three source regions. This experimental setup is illustrated in Fig. 2.

Observations of the tropospheric tracer HCN obtained from the Atmospheric Chemistry Experiment Fourier Transform

Spectrometer (ACE-FTS) satellite instrument (Bernath et al., 2005) are used to validate the artificial monsoon air mass tracer. These data have been presented and discussed by Randel et al. (2010) and Ploeger et al. (2017). Here, we use HCN from ACE-FTS level 2 data version 3.6 (Boone et al., 2005, 2013) to investigate the correspondence between pollutants in the stratosphere and transport via both monsoons. In addition, MLS version 4 retrievals of CO (Livesey et al., 2018) are used to investigate the influence of the monsoon anticyclones on the tracer distribution as shown in Fig. 1b, and MLS version 4 retrievals of

water vapor are used to diagnose vertical and horizontal transport of tracer from the monsoon regions and its influence on stratospheric composition.

## 3  Transport pathways into the stratosphere and influence on stratospheric composition

### 3.1  Zonal-mean perspective on transport

The evolution of the ASM tracer after its release is very similar to results shown by Ploeger et al. (2017, their Fig. 1). To avoid

redundancy, we only show the AOF distributions around ten months after the tracers are initialized (i.e. during April-June),

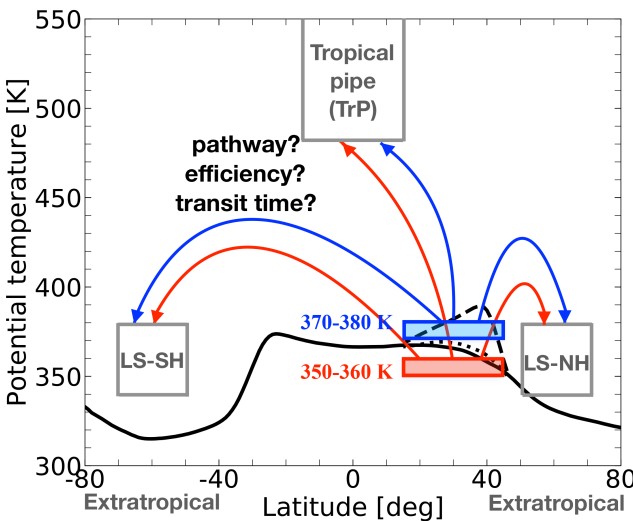

**Figure 2.** Definition of source regions (Box$_i$) at $350-360$ K (red) and $370-380$ K (blue). The following three destination regions (i.e. regions where transport statistics from the source regions are diagnosed) are considered: the tropical pipe (TrP: $15°$ S$-15°$ N), the extratropical lower stratosphere in the Southern Hemisphere (LS-SH: $50°$ S$-70°$ S), and the extratropical lower stratosphere in the Northern Hemisphere (LS-NH: $50°$ N$-70°$ N). The black dashed, dotted, and solid lines respectively mark lapse rate tropopause locations within the ASM region, the NASM region, and all other regions from ERA-Interim averaged over July$-$August 2010$-$2013.

focusing on tracers that have been transported into the global stratosphere. Figure 3 shows the average AOF from the ASM and NASM regions initialized in the $350-360$ K (Fig. 3a and Fig. 3b) and $370-380$ K (Fig. 3c and Fig. 3d) layers in July and August of 2010$-$2013 in the CLaMS-EI simulations during the following April$-$June. These results are based on four sets of CLaMS simulations, with the AOF set to 1 in the $350-360$ K or $370-380$ K layer over the ASM or NASM region on each day during July$-$August. Air is transported to the global stratosphere from both the ASM and the NASM. However, much more monsoon air originates from the ASM region than from the NASM region, and ASM air is transported to a higher level in the TrP by the following April$-$June when compared to NASM air. More monsoon tracer is transported into the stratosphere from the $350-360$ K layer than from the $370-380$ K layer for both the ASM and NASM regions. Monsoon air initialized at $350-360$ K is more prevalent in the SH stratosphere than in the NH, while monsoon air initialized at $370-380$ K is more likely to remain in the NH. The distribution of ASM tracer initialized at $370-380$ K (Fig. 3c) is similar to that reported by Ploeger et al. (2017, their Fig. 1d) based on a PV-barrier definition of the ASM, but the abundance of ASM air is greater in our results because our ASM domain is larger. The distribution of the monsoon tracer initialized at $340-350$ K (not shown) is also skewed toward the SH, similar to the results for $350-360$ K, while the distribution of the monsoon tracer from $360-370$ K is more symmetric between the hemispheres. In summary, during the following April$-$June, the monsoon tracers are more likely to end up in the SH stratosphere if they are initialized at lower levels ($340-350$ K or $350-360$ K), and more likely to end up

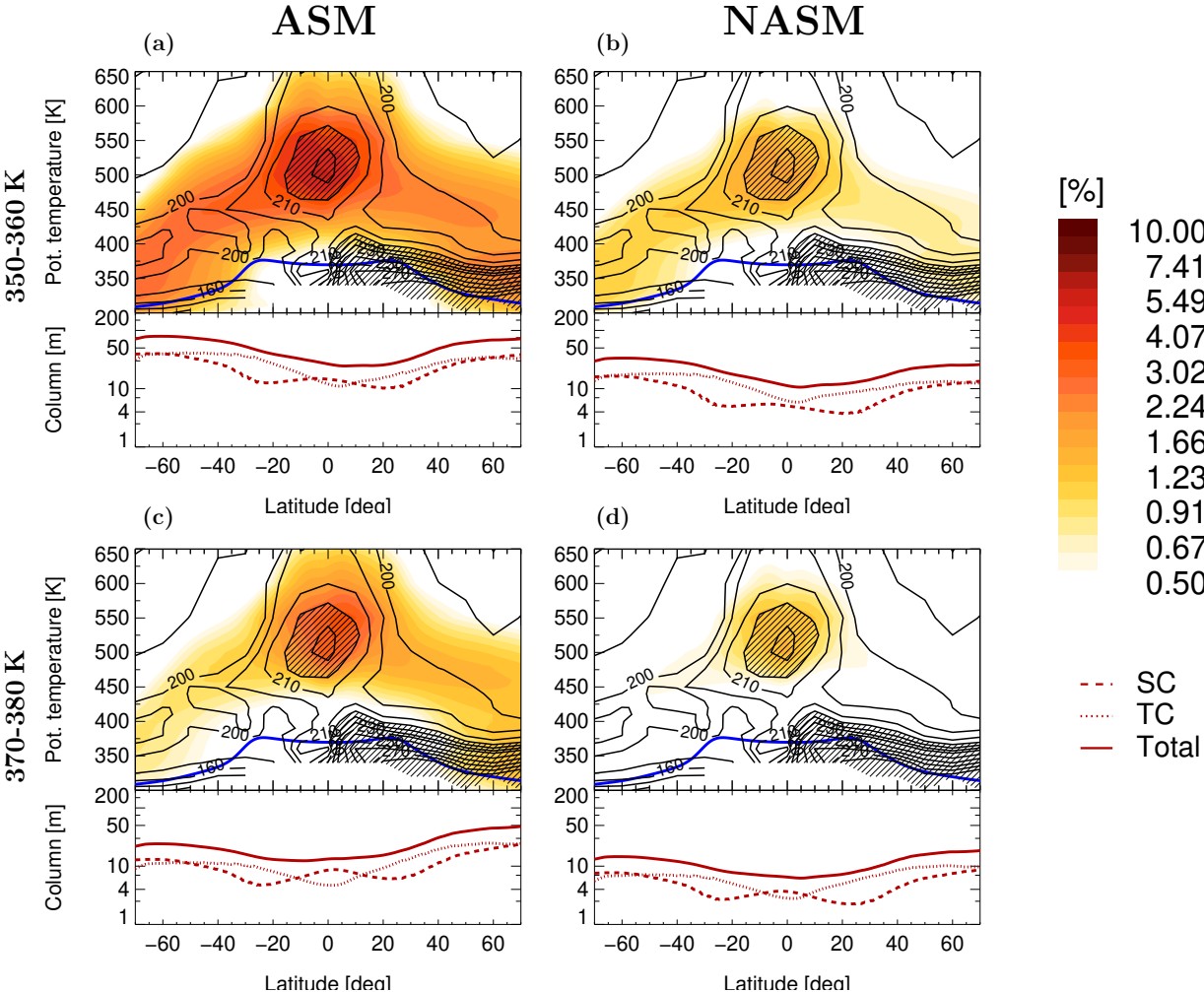

**Figure 3.** Climatological (2010−2013) zonal mean air mass origin fraction (AOF) from the ASM (a and c) and the NASM (b and d) initialized at 350−360 K (upper) and 370−380 K (lower) in July and August from CLaMS-EI (color shading) during the following April−June. HCN from ACE-FTS observations (black contours) is also shown. Regions with HCN volume mixing ratios greater than 215 pptv are hatched. Blue lines mark the lapse rate tropopause. The red lines show column integrals (thickness: in m) of the monsoon tracer above the tropopause (stratospheric column, SC: dashed), below the tropopause (tropospheric column, TC: dotted) and through the entire atmosphere (solid). Note the logarithmic color scale.

in the NH stratosphere if they are initialized at higher levels (360−370 K or 370−380 K). To avoid redundancy, we only show the results for tracers initialized in the 350−360 K and 370−380 K layers in this paper.

Ploeger et al. (2017) showed that the simulated ASM tracer based on driving CLaMS with ERA-Interim correlates well with ACE-FTS observations of HCN. Here, we find that the HCN observations show strong correlations with both the ASM and NASM tracers in the TrP. This result suggests that both monsoon regions could serve as sources of enhanced HCN to the stratosphere, with the ASM region representing a relatively strong source and the NASM region representing a relatively weak source. Although the HCN mixing ratio over the NASM is lower than that over the ASM, it is much higher than that over the tropics (Randel et al., 2010, their Fig. 1). The peak HCN mixing ratios in the TrP closely overlap with the peak contributions of monsoon tracers initialized in the 350−360 K layer, and are located slightly below the peak contributions of monsoon tracers initialized at 370−380 K. However, such vertical offsets should be interpreted with caution given likely overestimates of tropical upwelling in ERA-Interim (Dee et al., 2011; Wright and Fueglistaler, 2013; Ploeger et al., 2019; Tao et al., 2019).

Column integrals are calculated to further investigate the transport of the monsoon tracers. These integrals are defined as the thickness (in units of m) of pure monsoon air assuming standard temperature and pressure for each altitude. Column integrals of monsoon air above the tropopause (SC), below the tropopause (TC), and through the entire atmosphere are shown along the bottom of each panel in Fig. 3. The column integrals of ASM and NASM tracers show similar patterns in both hemispheres. The TC, SC, and total monsoon tracer columns initialized at 350−360 K show slightly larger values in the SH. The larger columns in the SH originating from the 350−360 K layer over the boreal monsoon regions may arise from a combination of three effects: weaker confinement of air inside the monsoon regions at this layer (Garny and Randel, 2016; Vogel et al., 2019), hemispheric differences in isentropic mixing (Orbe et al., 2013; Konopka et al., 2017), and the seasonality of the Brewer-Dobson circulation (Seviour et al., 2011; Konopka et al., 2015). A portion of the monsoon tracers initialized at the 350−360 K level enters the tropics due to wave-driven isentropic transport, ascends through the tropical tropopause layer (TTL), and is then advected into the SH through the shallow branch of the Brewer-Dobson circulation (Rosenlof, 1995; Konopka et al., 2015). By contrast, monsoon tracers released in the 370−380 K layer are more tightly confined to the NH, with higher column-integrated values for both the troposphere and stratosphere in the NH.

Figure 4 shows the AOF during the following April−June for the ASM and NASM tracers initialized at 350−360 K and 370−380 K in July−August based on CLaMS-M2 simulations. Similar to the results from CLaMS-EI, the fraction of ASM air is larger than that of NASM air in the TrP, and the ASM air reaches a higher vertical level. More monsoon tracer is transported into the stratosphere from lower levels than from higher levels, and the distributions of monsoon tracers also show hemispheric asymmetries similar to those in the CLaMS-EI case. However, despite these many similarities, the CLaMS-M2 simulations show much larger fractions of ASM and NASM air in the global stratosphere in comparison to those from CLaMS-EI. Another prominent difference concerns the vertical transport of monsoon air. The tracer peak from CLaMS-M2 is slightly below the HCN peak from satellite observations for tracers initialized at both the lower and the upper layers. This can be attributed to slower upwelling, which stems from smaller diabatic heating rates in the lower stratosphere in MERRA-2 (Ploeger et al., 2019; Tao et al., 2019). Slower upwelling leads to larger contributions from quasi-horizontal transport and smaller contributions from vertical transport relative to the CLaMS-EI simulations. Transport of tracers into the stratosphere takes a longer time in

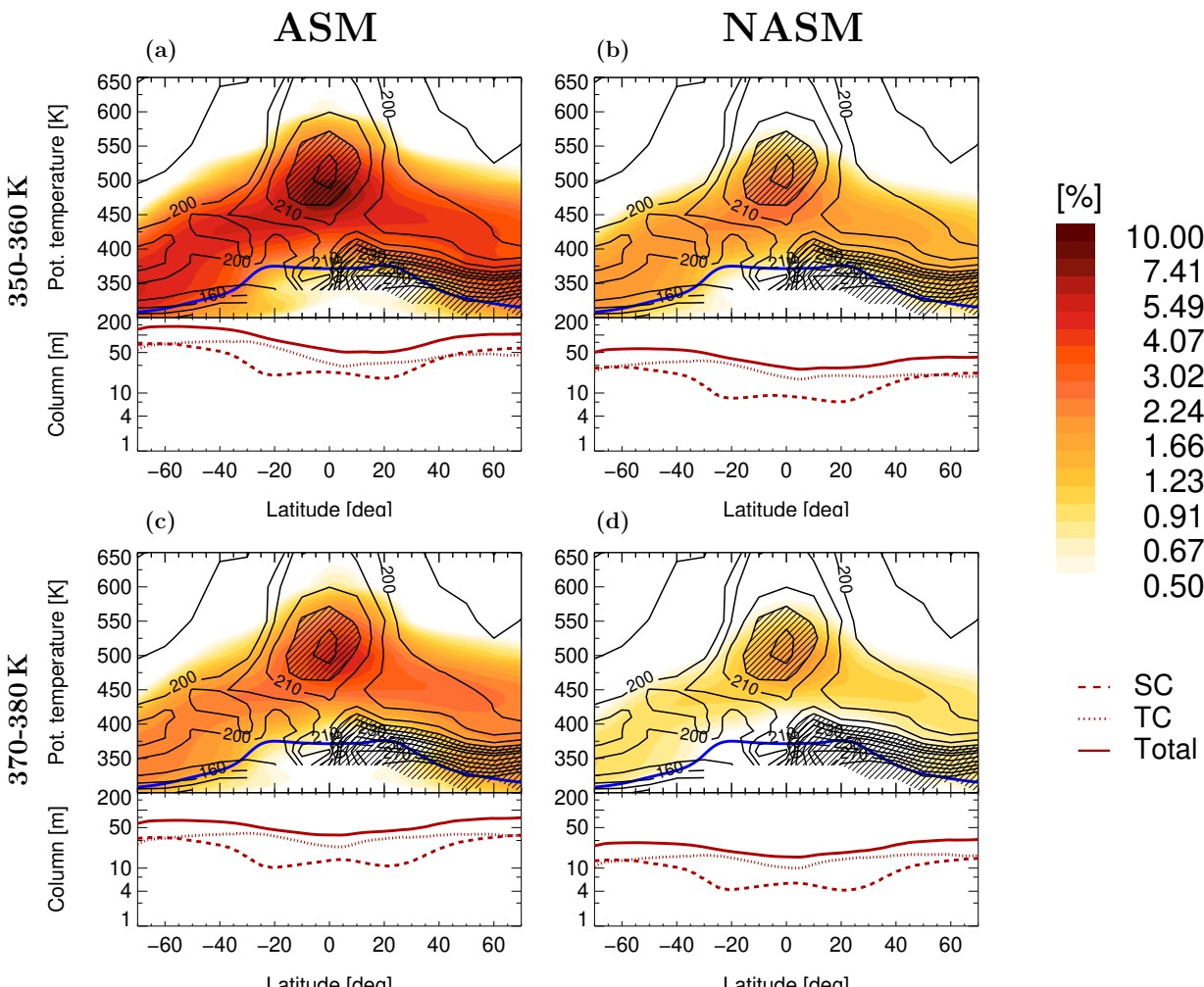

**Figure 4.** Same as Fig. 3 but for CLaMS-M2.

the CLaMS-M2 simulations, but a larger fraction of monsoon air reaches and remains in the stratosphere. Column-integrated monsoon tracer values from CLaMS-M2 are also much larger than those from CLaMS-EI. This difference arises in part because monsoon tracers from CLaMS-M2 reach relatively lower altitudes in the stratosphere by the following April−June compared to those from CLaMS-EI and air density is larger at lower altitudes. Differences between the SH and NH for monsoon tracers initialized in 370−380 K layer are smaller than those based on CLaMS-EI.

## 3.2 Two distinct transport pathways

In Sec. 3.1, we have discussed differences in transport into the stratosphere from the ASM and NASM regions, focusing on two different vertical levels of the monsoon source regions in the upper troposphere (the 350−360 K and 370−380 K layers). The differences in simulated transport of monsoon tracers released at different vertical levels imply the existence of multiple transport pathways from the monsoon source regions to the TrP. Along the first pathway, termed 'tropical pathway' in the following, the tracer is first horizontally advected to the tropics, where it ascends through the TTL into the TrP. Along the second pathway, termed 'monsoon pathway' in the following, the tracer is first lifted across the tropopause within the monsoon anticyclone and then transported isentropically to the tropical lower stratosphere and TrP. In this section, we endeavour to disentangle these two pathways and clarify the extent of transport from the monsoon source regions to the global stratosphere by separating the contributions of vertical transport within the tropics from those of horizontal transport into the tropics. We also compare the artificial tracers from the model simulations with MLS water vapor observations and CLaMS-simulated water vapor to investigate the influence of the monsoons on stratospheric chemical composition.

The vertical transport of ASM and NASM air into the tropical stratosphere ($15° \text{S} − 15° \text{N}$) in CLaMS-EI is illustrated using the familiar 'tape-recorder' form (Mote et al., 1996) in Fig. 5. The transport barrier between the tropics and subtropics is weaker during boreal summer (Chen, 1995; Haynes and Shuckburgh, 2000). Monsoon tracers released at lower levels are especially abundant in the tropical UTLS during summer, indicating that a large fraction of the monsoon air from 350−360 K reaching the TrP may be advected quasi-horizontally into the tropics before ascending, in agreement with previous results based on trajectories initialized at 360 K (Garny and Randel, 2016). We also find that less of the monsoon tracer reaches the tropics when it is initialized at higher levels relative to when it is initialized at lower levels. This difference suggests that tracers initialized at higher levels within the monsoon regions primarily ascend locally before being mixed or advected into the tropics. ASM air rises slightly faster than NASM air and reaches slightly higher altitudes by the end of the simulation period. These features are consistent with the results discussed above.

MLS water vapor observations are included for additional context in Fig. 5. The annual mean of MLS water vapor is removed to highlight the annually-repeating water vapor tape recorder. The simulated ASM and NASM tracers correlate well with the 'wet' phase of the tape recorder (positive anomalies after removing the annual mean), which starts each year in boreal summer. The close correspondence between the monsoon air mass tracers and the 'wet' phase of the water vapor tape recorder in the tropics is consistent with the idea that the ASM and NASM contribute to moistening the tropical lower stratosphere during boreal summer (e.g. Dethof et al., 1999; Bannister et al., 2004; Fu et al., 2006; James et al., 2008; Randel et al., 2012). Note that large concentrations of monsoon tracers in the tropics do not always correlate with large water vapor mixing ratios in

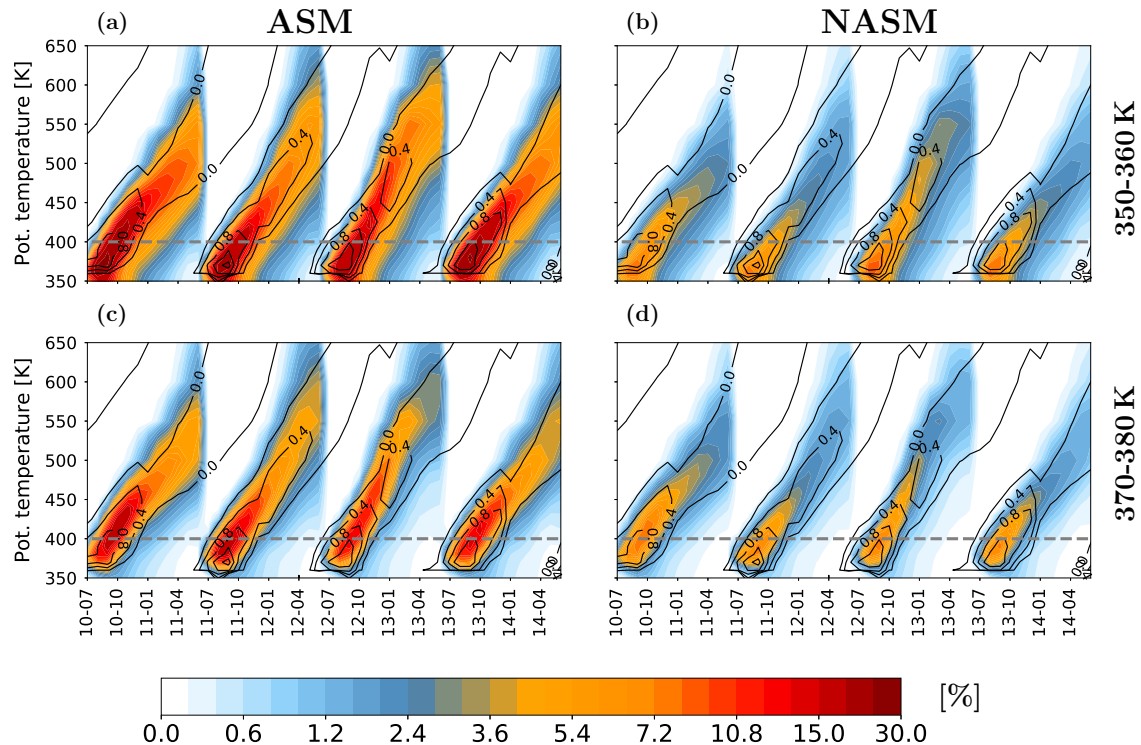

**Figure 5.** Potential temperature−time sections during July 2010 to June 2014 of zonal-mean monsoon air mass origin fraction (AOF: color shading) between 15° S and 15° N released in the 350−360 K (upper; a and b) and 370−380 K (lower; c and d) layers within the ASM (left; a and c) and NASM (right; b and d) based on CLaMS-EI simulations. Tape-recorder signals calculated from Aura MLS water vapor (in ppmv) with the annual average from each year removed are shown as black contours for context. Grey dashed lines mark the 400 K level. Note that the tracer is set to zero everywhere on 1 July of each year.

the tropical UTLS, as water vapor content at these altitudes depends not only on the origin of the air parcel but also on its temperature history (Fueglistaler and Haynes, 2005; Nützel et al., 2019).

CLaMS-M2 simulations (not shown) have much in common with the CLaMS-EI results with respect to the transport of monsoon tracers into the tropics. However, there are also some differences. As mentioned above, the vertical transport of the ASM tracers is slightly weaker, with the tracers reaching slightly lower altitudes by the end of the simulation period.

Likely linked to this slower upwelling, more monsoon air arrives in the tropics between each summer monsoon season and the following spring compared to CLaMS-EI simulations. A relatively large fraction of monsoon air is transported quasi-horizontally into the tropics before ascending to the TrP. The slower upwelling in MERRA-2 also manifests in a slight lag between the simulated monsoon tracer distribution and the MLS water vapor tape recorder signal.

To investigate the influence of transport from the boreal monsoon regions on the lowermost stratosphere, we examine the hor-

izontal spread of the ASM and NASM tracers on the 400 K isentropic surface based on the CLaMS-EI simulations (Figure 6).

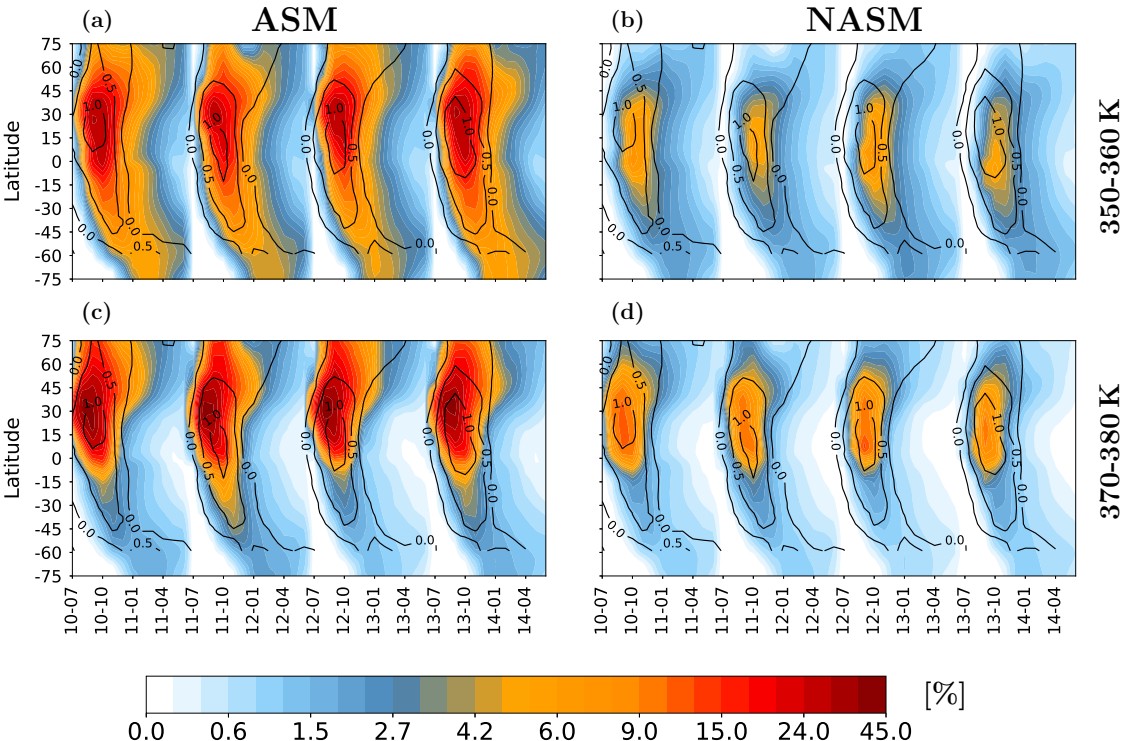

**Figure 6.** Zonal-mean tracer concentrations (color shading) for the ASM (left; a and c) and NASM (right; b and d) tracers initialized at 350−360 K (upper; a and b) and 370−380 K (lower; c and d) as functions of time and latitude on the 400 K isentropic surface based on CLaMS-EI (see also Fig. 5). MLS water vapor retrievals (in ppmv) with the annual average from each year removed are shown as black contours for context. Note that the tracer is set to zero everywhere on 1 July of each year.

Tracers initialized on both levels show broadly similar features at 400 K, although the ASM contributes much more to the lower stratosphere than does the NASM. The ASM and NASM tracers initialized at 350−360 K are less confined in latitude, with much more transport to the SH than tracers initialized at 370−380 K. Tracers initialized at 370−380 K are more confined to the monsoon region and the NH, consistent with trajectory simulations initialized at 380 K (Garny and Randel, 2016). The ASM and NASM tracers coincide well with the MLS water vapor 'wet' signal in the lateral tape recorder, which is again consistent with the idea that the monsoons contribute substantially to water vapor concentrations in the stratosphere. The monsoon tracer distributions based on CLaMS-M2 are more widespread than those based on CLaMS-EI, with a greater tendency to remain in the extratropical stratosphere regardless of the initialization level. Both sets of simulations produce close matches between the ASM and NASM tracer peaks and the 'wet' signal in MLS water vapor, indicating that horizontal transport in CLaMS-M2 is in good agreement with that in CLaMS-EI.

The qualitative correspondence between the monsoon tracers and the 'wet' phases of the vertical and horizontal tape recorders suggests a strong connection between stratospheric water and transport from the monsoon regions. To further in-

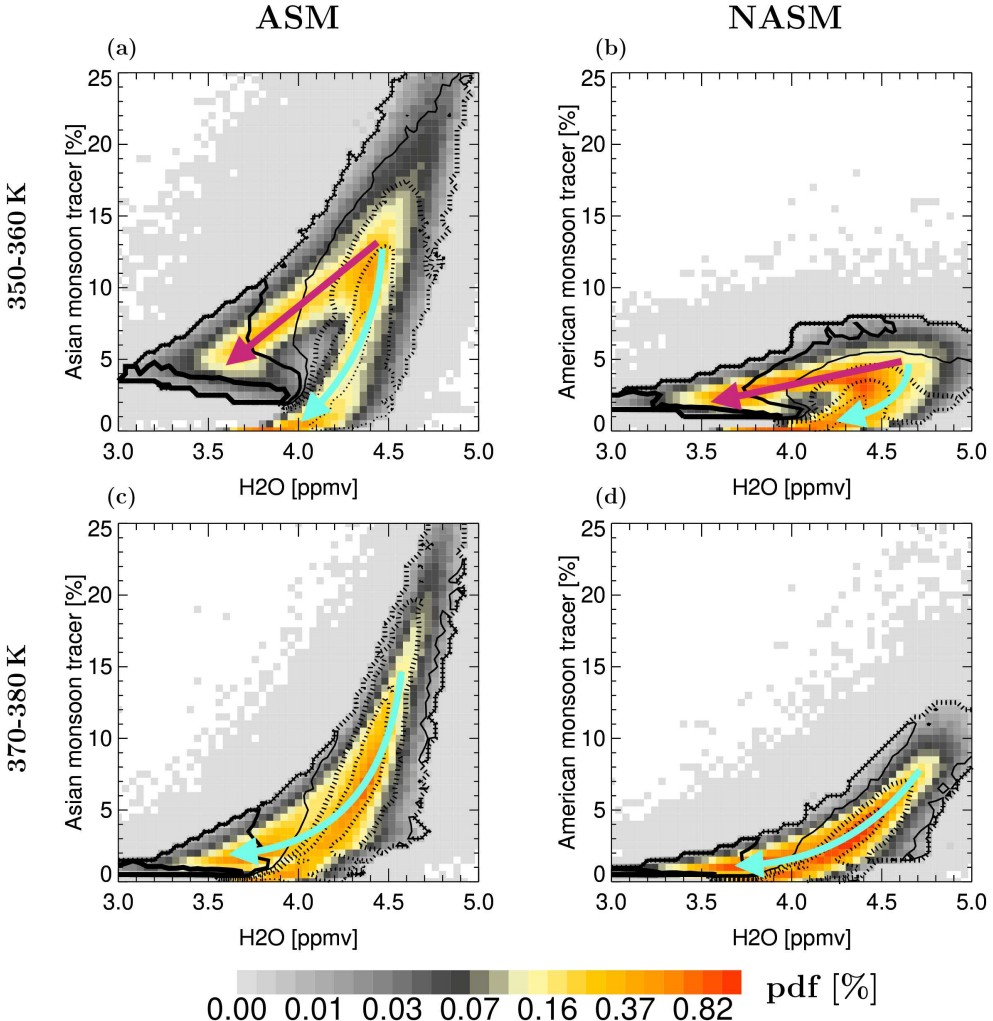

**Figure 7.** Correlations between CLaMS-simulated water vapor and the ASM (left; a and c) and NASM (right; b and d) tracers in the tropical stratosphere ($15°$ S$-15°$ N; $420-500$ K) during from November 2010 to May 2011 with monsoon tracers released in the $350-360$ K (upper; a and b) and $370-380$ K (lower; c and d) layers based on CLaMS-EI. Contours mark young (less than nine months) air mass fractions of 24%, 33%, and 42% (from thin to thick dotted isolines) and 51%, 60%, and 69% (from thin to thick solid isolines). The cyan and magenta arrows respectively show the monsoon pathway and tropical pathway.

vestigate the extent to which the monsoon tracers can explain variability in lower stratospheric water vapor and clarify the differences between the tropical pathway and the monsoon pathway into the TrP, we correlate the monsoon tracer amounts with CLaMS-simulated concentrations of stratospheric water vapor, which have been successfully validated and used in many previous studies (e.g. Vogel et al., 2016; Rolf et al., 2018; Tao et al., 2019, and references therein). We analyse the data from November to May, i.e. about $4-10$ months after the release of the monsoon tracers (as discussed below, this is the earliest that the tracers may reach the TrP). As shown in Fig. 7, a strong positive correlation exists between the monsoon tracers and water vapor in the tropics. Larger ASM and NASM air fractions correspond to larger water vapor mixing ratios in the tropical lower stratosphere ($420-500\,\text{K}$). The distinct differences between the upper panels (tracers released in the $350-360\,\text{K}$ layer) and the lower panels (tracers released in the $370-380\,\text{K}$ layer) reflect the two transport pathways from the ASM and NASM to the tropical lower stratosphere. The right branch in Fig. 7a (Fig. 7b) mimics the joint distribution of ASM (NASM) tracers from $370-380\,\text{K}$ and water vapor shown in Fig. 7c (Fig. 7d). This branch thus indicates the correlation along the monsoon transport pathway (cyan arrow), in which air ascends locally across the tropopause and then enters the tropics laterally. The left branch in Fig. 7a (Fig. 7b) is then linked to the correlation along the tropical pathway (magenta arrow), in which air from the upper troposphere above the ASM (NASM) region enters the tropics quasi-horizontally before ascending via tropical upwelling.

The tropical pathway is more common for tracers released at $350-360\,\text{K}$. Most of the monsoon tracers released at $370-380\,\text{K}$ are transported to the TrP along the monsoon pathway, although the tropical pathway is not completely absent for tracers released in this layer. The monsoon pathway coincides well with the upward spiraling range highlighted by Vogel et al. (2019). The water vapor mixing ratio along the tropical pathway is lower than that along the monsoon pathway because more dehydration is caused by the lower tropopause temperature over the tropics (figure omitted). The mass fraction of air younger than nine months is also shown in Fig. 7 (black contours) to further evaluate the vertical components of the pathways by which monsoon air reaches the tropical lower stratosphere. Young air mass fractions are considerably larger for the tropical pathway, suggesting that overall ascent rates over the tropics (along the tropical pathway) are much faster than those over the monsoon regions (along the monsoon pathway) until the next early summer. Here, a single one-year simulation is used for each tracer because interannual variability obscures the structure in the correlation between water vapor and monsoon tracers. Results from other years are similar (not shown).

## 4   Transport budget

In this section, we investigate the total budget (in terms of the AOF) and the efficiencies of transport from different layers over the monsoon source regions into the global stratosphere. Three destination regions as defined in Fig. 2 are of particular interest: the tropical pipe (TrP), the extratropical lower stratosphere in the NH (LS-NH) and the extratropical lower stratosphere in the SH (LS-SH). Because transport across the TTL is generally regarded as the 'main gateway into the stratosphere' (Fueglistaler et al., 2009), we compare the ASM- and NASM-related transport metrics with corresponding quantities calculated for a pure tropical source region ($15°\,\text{S}-15°\,\text{N}$).

## 4.1 Time evolution of the monsoon tracer

Figure 8a shows CLaMS-EI time series of the ASM tracer in the three destination regions (TrP, LS-NH and LS-SH) defined in Fig. 2. The ASM tracer initialized at $370-380\,K$ shows the largest air fraction in the LS-NH in the middle of September, with a peak value around 22%. The peak date is earlier than that shown by (Ploeger et al., 2017, their Fig. 2) (peak around 15%). This is because the geographical domain of the ASM tracer used here is larger than that based on the PV barrier defintion used by Ploeger et al. (2017). As a consequence, air parcels that are not well confined inside the anticyclone may reach the LS-NH faster. The ASM tracers released in the $350-360\,K$ layer show two peaks in the LS-NH. The first of these is at the beginning of September and is related to rapid isentropic poleward transport. The second occurs about one month later relative to the tracers initialized in the $370-380\,K$ layer due to the need for additional (spiraling) ascent within the anticyclone (Ploeger et al., 2010; Garny and Randel, 2016; Vogel et al., 2019). During the first three months, fewer ASM tracers released at $350-360\,K$ are diagnosed in the LS-NH relative to ASM tracers released at $370-380\,K$. However, after three months the ASM tracers released at $350-360\,K$ contribute significantly more to the composition in the LS-NH and remain in the LS-NH for longer when compared to tracers released at $370-380\,K$. Similar transitions in the relative concentrations of tracers released at $350-360\,K$ and $370-380\,K$ can be diagnosed in the TrP region, although the maxima are about six months later compared to those in the LS-NH. The ASM tracers start to reach the TrP region around November, peaking in the TrP around March at a value $\sim4\%$ for tracers released in the $370-380\,K$ layer. The timing of this peak is about half a month earlier than that of ASM tracers released from $350-360\,K$, although the latter has a larger peak value ($\sim7\%$). Tracers released at $350-360\,K$ are much more abundant in the LS-SH than tracers released at $370-380\,K$ throughout the year.

The corresponding time series of air fractions from the NASM source region to the three destination regions are illustrated in Fig. 8b. These time series show some important similarities to those based on the ASM tracers. The NASM tracers initialized in both the $350-360\,K$ and $370-380\,K$ layers reach their peak values in the LS-NH about two weeks earlier than the corresponding ASM tracer concentrations, likely owing to the weaker dynamical confinement of NASM air in the upper troposphere (see also Fig. 1d). The amount of NASM tracer initialized at $370-380\,K$ that reaches the LS-NH (21%) is similar to the amount of ASM tracer initialized at the same vertical layer that reaches the LS-NH. This similarity is also manifest in the comparable CO concentrations along $90°\,E$ and $90°\,W$ in the LS-NH (Fig. 1c and Fig. 1d). The amount of NASM tracer initialized at $350-360\,K$ is much less, in part because the 370-380 K layer is above the tropopause in the NASM region. However, NASM air fractions in the LS-SH and TrP are much smaller than the corresponding ASM air fractions in these regions, generally by more than a factor of 2. As for the ASM, NASM tracers initialized at $350-360\,K$ are more abundant in the three destination regions by the end relative to NASM tracers initialized at $370-380\,K$.

Most tropospheric air that is transported into the global stratosphere passes through the TTL via tropical upwelling. Thus, to evaluate the relative contributions of the ASM and NASM sources to the stratosphere, we use the transport of air originating in the same vertical layers over the tropics ($15°\,S - 15°\,N$) as a benchmark. By comparing tracers tracking air of tropical origin in the three destination regions (Fig. 8c) with those tracking air with monsoon origin in the same destination regions, we conclude that ASM tracers are as abundant as tropical tracers in the LS-NH. Moreover, the abundance of the NASM tracer

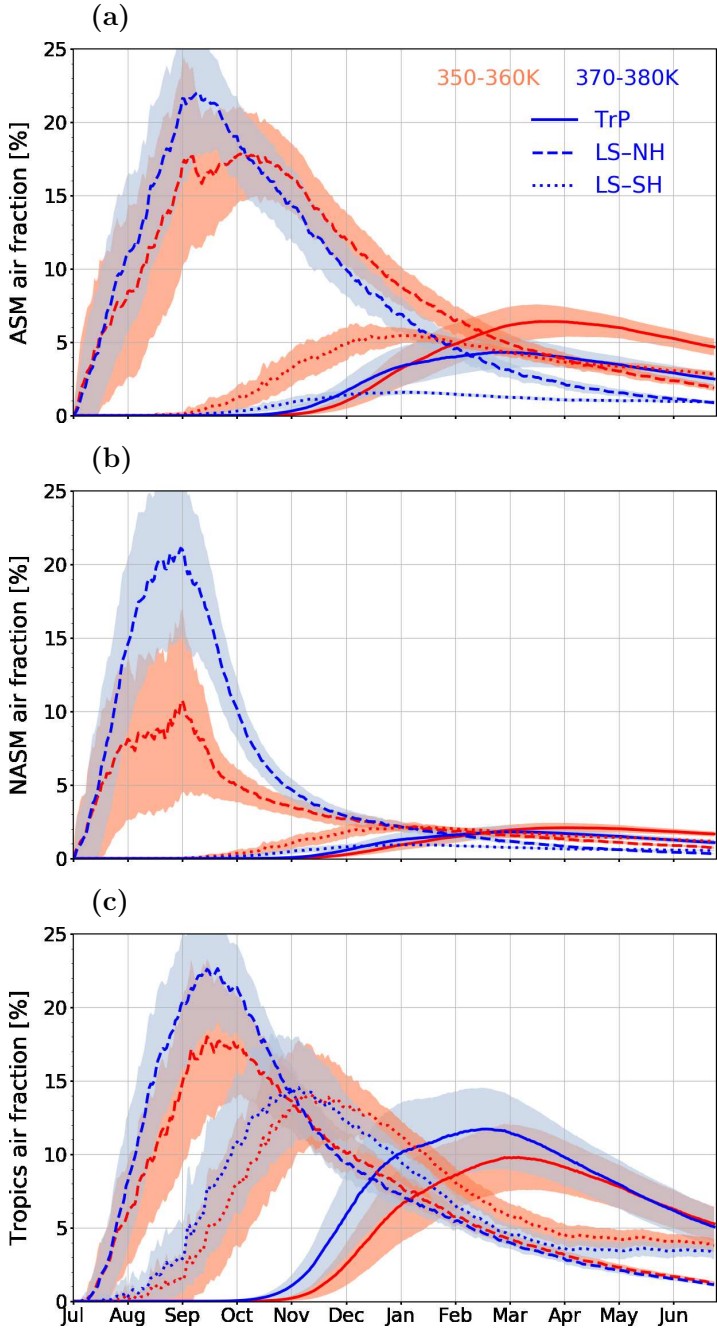

**Figure 8.** Climatological time series based on CLaMS-EI simulations of air mass origin fractions (AOFs, in %) from the three source regions: (a) ASM, (b) NASM and (c) tropics as diagnosed in three destination regions (see Fig. 2): tropical pipe (TrP, solid line), extratropical lower stratosphere in the NH (LS-NH, dashed line) and in the SH (LS-SH, dotted line). Shading shows the mean standard deviation in the zonal average (multiplied by 0.5 for better visibility). Red and blue lines respectively represent the tracers released in the $350-360$ K and $370-380$ K layer.

| ERA-Interim | 340−350 K | 350−360 K | 360−370 K | 370−380 K |
|---|---|---|---|---|
| ASM | $2.276\times10^{10}$ | $1.877\times10^{10}$ | $1.179\times10^{10}$ | $6.118\times10^{9}$ |
| NASM | $1.916\times10^{10}$ | $1.042\times10^{10}$ | $5.649\times10^{9}$ | $3.930\times10^{9}$ |
| Tropics | $1.188\times10^{11}$ | $5.950\times10^{10}$ | $2.743\times10^{10}$ | $1.621\times10^{10}$ |
| MERRA-2 | 340−350 K | 350−360 K | 360−370 K | 370−380 K |
| ASM | $2.284\times10^{10}$ | $1.890\times10^{10}$ | $1.124\times10^{10}$ | $6.334\times10^{9}$ |
| NASM | $2.006\times10^{10}$ | $9.868\times10^{9}$ | $5.927\times10^{9}$ | $3.993\times10^{9}$ |
| Tropics | $1.286\times10^{11}$ | $5.465\times10^{10}$ | $2.908\times10^{10}$ | $1.652\times10^{10}$ |

**Table 1.** Typical masses of air contained in each box for the test experiments over the ASM, NASM and tropical source regions in CLaMS-EI (top) and CLaMS-M2 (bottom). All masses are reported in kg.

initialized in the 370−380 K layer is also comparable to the abundances of ASM or tropical tracers in the LS-NH region. Inside the TrP, the peak ASM contribution is smaller than the peak tropical contribution in early spring. However, the abundance of ASM air originating from 350−360 K in the TrP is comparable to that from the tropics at the end of the simulation period. Both values are about three times larger than the NASM contribution. Much more air in the LS-SH originates from the tropical upper troposphere than from the two monsoon regions.

The time series of all source air fractions discussed above calculated from CLaMS-M2 are very similar to those derived from CLaMS-EI (not shown). Even daily perturbations of such tracers driven by Rossby waves and diagnosed on individual isentropic surfaces show very strong similarities, as outlined later. ASM contributions based on CLaMS-M2 lag those based on CLaMS-EI by about one month, first reaching the TrP in December rather than November. This delay is related to the slower tropical upwelling in MERRA-2 relative to ERA-Interim as discussed above. For the same reason, CLaMS-M2 produces larger amounts of ASM tracers in the stratosphere at the end of the simulation period. The time series of NASM tracers based on CLaMS-M2 are likewise similar to those based on CLaMS-EI, again including daily variability. For the tropical source, CLaMS-M2 shows similar peak values to CLaMS-EI in the LS-SH destination region, but lower peak values in the LS-NH and TrP. However, the amount of air of tropical origin reaching the TrP is larger in CLaMS-M2 than in CLaMS-EI starting from April.

## 4.2 Transport efficiency

Our comparison of AOFs conditioned on different transport pathways is partially hampered by the fact that the source regions have different masses. To illustrate such differences in the domains, Table 1 shows the mass calculated for all considered source regions (ASM, NASM and tropics) based on both reanalyses. To better control for the potential influence of differences in the

sizes of the source domains on our quantitative estimates of transport to the stratosphere, we define the efficiency of transport (TE) using the air mass of the source and destination domains, $m_{\text{source}}$ and $m_{\text{dest}}$, to normalize the mass transported from each

345 source region to the defined stratospheric destination regions. Thus, TE is defined as AOF $\times (m_{\text{dest}}/m_{\text{source}})$.

Figure 9a compares the TE from the three source regions (ASM, NASM and Tropics) each divided into two layers ($350-360\,\text{K}$ and $370-380\,\text{K}$) to the LS-NH destination region. In contrast to the total budget, where tropical and monsoon contributions are very similar (Fig. 8), the ASM/NASM monsoons dominate the TE into the LS-NH, especially for sources in the $370-380\,\text{K}$ altitude range. At the beginning, the TE is larger for the NASM than for the ASM; the larger NASM TE is in part related to the

350 fact that the $370-380\,\text{K}$ layer is above the tropopause over the NASM region and the NASM tracers are less confined. However, transport from the ASM becomes most efficient starting from October. Perhaps more surprising, the TE into the LS-SH is also dominated by the ASM and NASM regions for tracers released at $350-360\,\text{K}$ after a certain date (see Fig. 9b). For the first $4-5$ months, the TE is larger from the tropics than from the monsoon regions, especially for tracers released between $370\,\text{K}$ and $380\,\text{K}$. However, transport from the $350-360\,\text{K}$ layers in the ASM and NASM regions reaches the LS-SH more efficiently

than transport from the same layer in the tropics starting from December or January (with TE from the ASM larger than that from the NASM). The time evolution of the source-resolved TE into the TrP is shown in Fig. 9c. Here, the highest TE values are along the pathways starting in the ASM source region after some certain date, followed by those starting in the tropical and NASM source regions. The NASM TE exceeds the tropical TE by March for tracers from $350-360\,\text{K}$. In contrast to the total source budget (Fig. 8), transport is consistently less efficient from lower levels than from higher levels.

Patterns of TE based on CLaMS-M2 are similar to those based on CLaMS-EI (not shown), although there are a few important differences. First, TEs from the ASM and NASM source regions to the LS-SH destination region are much larger in CLaMS-M2 than in CLaMS-EI. Both values exceed those from the tropical source region by December (for tracers initialized at $350-360\,\text{K}$) or February (for tracers initialized at $370-380\,\text{K}$). Second, the TE to the TrP in CLaMS-M2 shows slightly higher values for both the ASM and NASM sources and lower values for tropical upwelling relative to CLaMS-EI. TE from the

NASM to the TrP slightly exceeds that from the tropics to the TrP starting in March (April) for tracers initialized at $350-360\,\text{K}$ ($370-380\,\text{K}$).

## 5   Discussion

The CLaMS-EI and CLaMS-M2 simulations both show that two major pathways, here referred to as the monsoon and tropical, connect the upper troposphere in the ASM and NASM regions with the TrP. We have further shown that the AOF from the

370 ASM upper troposphere ($350-360\,\text{K}$) to the TrP is comparable to the AOF from the tropical upper troposphere ten months after the tracers are released. The most efficient transport to the TrP among the analyzed source regions is that from the ASM at $370-380\,\text{K}$. These robust characteristics of transport in the CLaMS-EI and CLaMS-M2 simulations are based on zonal-mean metrics.

To further investigate the sensitivity of the results to the choice of reanalysis, zonally-resolved distributions of ASM tracers

initialized in the $370-380\,\text{K}$ layer are shown for the $380\,\text{K}$ isentropic surface on 24 August 2012 based on CLaMS-EI and

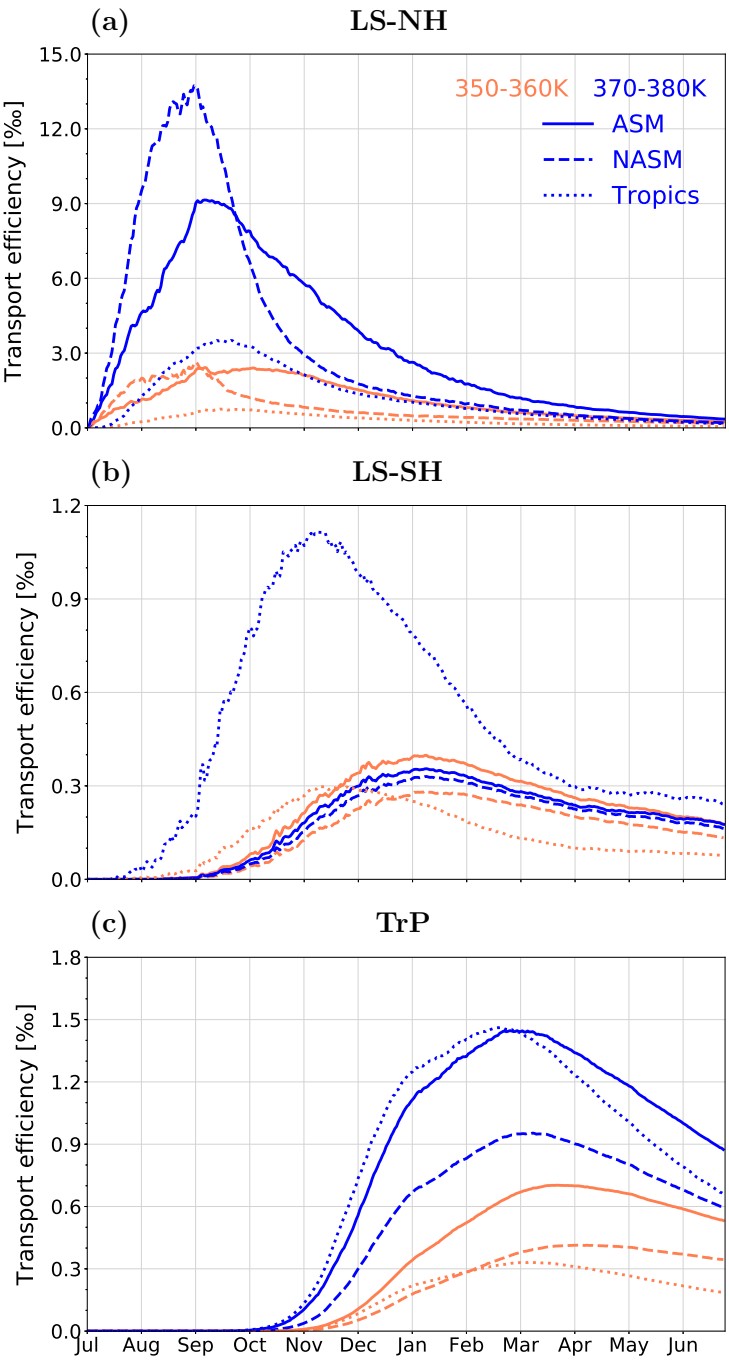

**Figure 9.** Climatological time series of transport efficiency (TE) as derived from the CLaMS-EI simulations and calculated for transport from the ASM (solid line), NASM (dashed line) and tropics (dotted line) source regions to the (a) LS-NH , (b) LS-SH and (c) TrP destination regions. TE (in ‰) is defined as AOF $\times$ ($m_{\mathrm{dest}}/m_{\mathrm{source}}$), where $m_{\mathrm{source}}$ and $m_{\mathrm{dest}}$ are the masses of the source and destination regions, respectively. Different colors of the lines represent tracers released at different levels.

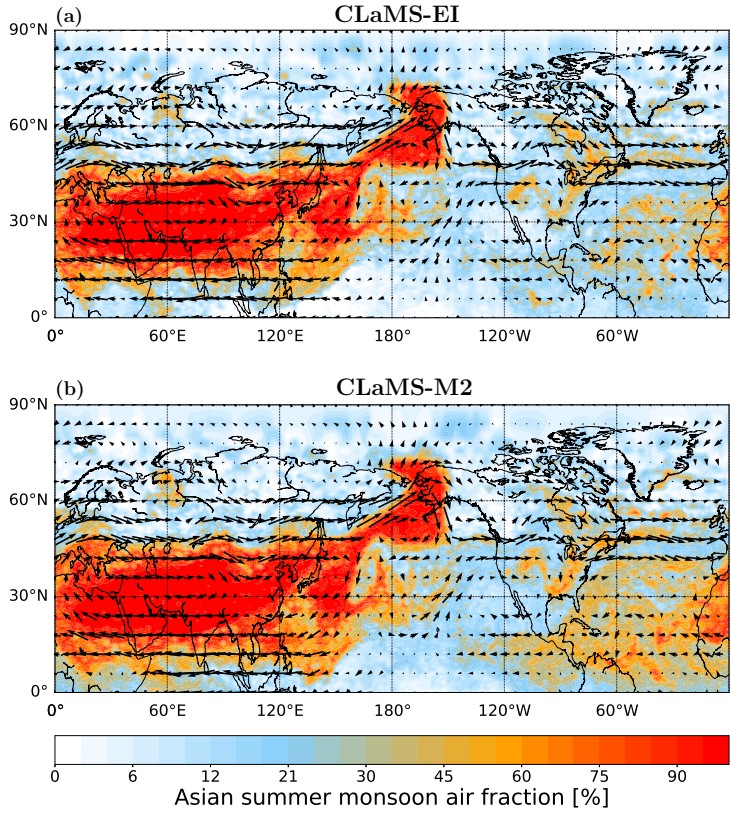

**Figure 10.** A snapshot of the horizontal distribution of the ASM tracer initialized in the $370-380$ K layer on the 380 K isentropic surface on 24 August 2012 based on the (a) CLaMS-EI and (b) CLaMS-M2 simulations. Arrows show horizontal wind on the same isentropic surface.

CLaMS-M2 (Fig. 10). The ASM tracer is transported as far north as 75° N by eddies that detach from the ASM anticyclone. This type of feature is associated with Rossby wave breaking and can cause large fluctuations of monsoon air fraction in the LS-NH region (e.g. Popovic and Plumb, 2001; Ploeger et al., 2013; Vogel et al., 2014; Ungermann et al., 2016; Fadnavis et al., 2018). The consistency of CLaMS-EI and CLaMS-M2 suggests good agreement between these two simulations with respect to horizontal transport (see also Sec. 3.2). CLaMS-EI and CLaMS-M2 also show good agreement with respect to the horizontal distributions of monsoon tracers on other days (not shown). We select this example because it represents a clear episode of Rossby wave breaking and eastward eddy shedding.

Despite many similarities in the simulated transport between CLaMS-EI and CLaMS-M2, there are also significant differences. As discussed recently by Tao et al. (2019) and Ploeger et al. (2019), radiative heating rates in the TTL and lower stratosphere are substantially larger in ERA-Interim than in MERRA-2. This difference directly affects the tropical upwelling as represented in CLaMS. We have shown that the monsoon tracers reach different maximum altitudes in the TrP during April$-$June in these two simulations. The stronger tropical upwelling in CLaMS-EI lifts monsoon air to higher altitudes more

quickly than the weaker tropical upwelling in CLaMS-M2. Accordingly, the transit time to a given altitude in the TrP is slightly shorter in CLaMS-EI than in CLaMS-M2. Moreover, the smaller magnitudes of heating rates (both positive and negative) in CLaMS-M2 lead to monsoon air accumulating in the lowermost stratosphere and TrP (Fig. 4). Monsoon tracers thus retain high values in the stratosphere for longer in the CLaMS-M2 simulations. Slower cross-isentropic transport makes quasi-horizontal (isentropic) transport into the tropics more effective.

Another open question is the sensitivity of our results to the geometric definition of the source regions or to the length of the period during which monsoon tracers are initialized. To address this, we have conducted simulations with modified definitions of the ASM domain (by shifting the southern boundary to the north from $15°$ N to $20°$ N) and NASM domain (by shifting the western boundary to the east from $160°$ W to $120°$ W). The resulting tracer distribution patterns are very similar. Owing to the smaller sizes of the source domains, AOFs in the destination regions are smaller in these test cases than in the reference case, especially for transport from the NASM to the LS-NH. However, differences in TE are much less pronounced. This latter result is consistent with the definition of TE, which accounts for the sizes of the source and destination regions. The insensitivity of the results to the source domain boundary in our study is in agreement with the findings shown by Garny and Randel (2016), where they found that the differences are small regarding a change in the source region definition from PV-contour to box. If monsoon tracers are initialized from 15 June to 15 September instead of only from 1 July to 31 August, the abundances of monsoon air in the stratosphere are larger compared to the reference case. However, the transport patterns are very similar, including the hemispheric asymmetry in the distribution of monsoon tracers and the largest TE being associated with transport from the $370-380$ K layer above the ASM. Hence, the conclusions of this study are qualitatively robust to reasonable definitions of the monsoon tracers.

Figure 11 provides a schematic illustration of the main transport pathways from the ASM (Fig. 11a) and NASM (Fig. 11b) source regions to the three stratospheric destination regions. The average AOFs, TEs and transit times associated with these pathways are also listed based on the CLaMS-EI results. The largest contribution, representing about 22% (18%) of the air in the LS-NH, consists of ASM tracers initialized at $370-380$ K ($350-360$ K), with an overall TE of 9‰ (2.5‰) and a transit time of ∼3 months. The contribution of the NASM tracers initialized at $370-380$ K to the LS-NH is comparable to that of the corresponding ASM tracers, while the contribution of the NASM tracers initialized at $350-360$ K is much smaller than that of the ASM tracers initialized in the same layer. Although not shown here, these three quantitative transport metrics are quite similar to CLaMS-M2 results with respect to transport from the ASM and NASM regions to the LS-NH. This similarity confirms our finding that these two reanalyses provide very consistent representations of horizontal transport. A maximum of 7% of the air arrives in the TrP via the two main transport pathways (monsoon and tropical) from the $350-360$ K layer in the ASM region. The transit time is about 8.5 months, with a peak efficiency of 0.7‰. The ASM tracers initialized at $370-380$ K take about 8 months to arrive in the TrP. Although only about 4% of the air in the TrP comes from the $370-380$ K layer above the ASM, the peak transport efficiency (1.45‰) is approximately twice as high as that from the $350-360$ K layer. Transport from source regions above the NASM contributes much less to air in the TrP, with slightly longer transit times compared to transport from the ASM. The maximum contribution of the NASM is larger from the $350-360$ K layer (2.5%) than from the $370-380$ K layer (2%), but with a lower peak efficiency from lower layer (0.4‰ relative to 0.95‰).

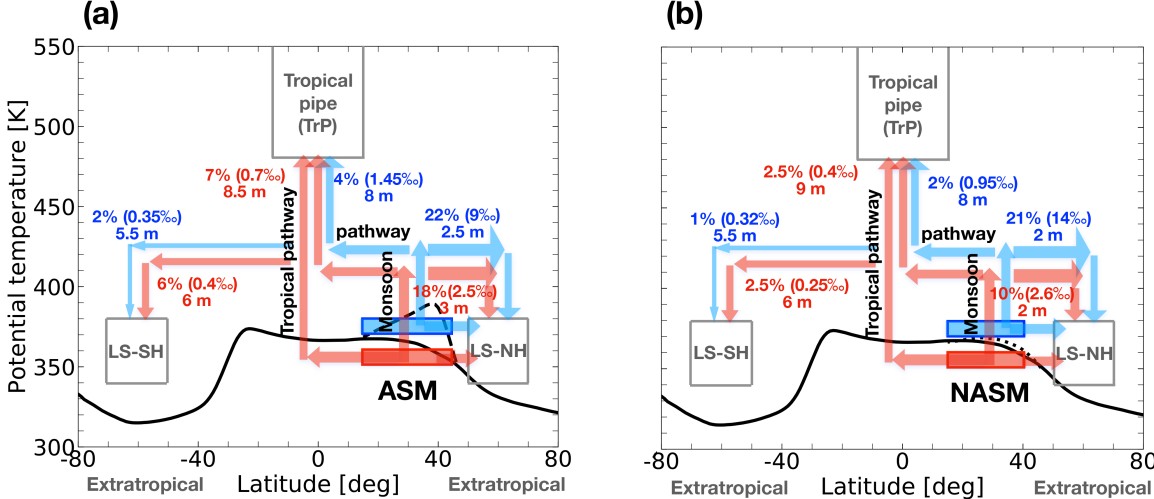

**Figure 11.** Schematic of tracer transport from the (a) ASM and (b) NASM region into the TrP, LS-NH and LS-SH. The red and blue boxes respectively represent the tracers initialized in the $350-360\,\mathrm{K}$ and $370-380\,\mathrm{K}$ layers. The arrows illustrate the dominant transport pathways. The numbers quantify the maximum AOF (in %), the maximum transport efficiency (TE, in ‰) and the mean transit time (in month) from the monsoon regions to three destination regions based on the CLaMS-EI simulations. The black dashed, dotted, and solid lines respectively mark lapse rate tropopause locations within the ASM region, the NASM region, and all other regions from ERA-Interim averaged over July−August 2010−2013.

Transit times from the monsoon regions to the TrP are about one month longer in the CLaMS-M2 simulations than in the ClaMS-EI simulations (not shown). The overall contributions and efficiencies of transport from the ASM and NASM to the
TrP are also slightly larger in CLaMS-M2 than those in CLaMS-EI. The contributions from the ASM and NASM regions to the LS-SH are significantly different between tracers initialized at $350-360\,\mathrm{K}$ and tracers initialized at $370-380\,\mathrm{K}$. Moreover, the absolute contributions based on CLaMS-M2 are much larger, with higher transport efficiencies and slightly longer transit times relative to CLaMS-EI. Since the tracers in the simulations are reset every July, we cannot infer how differences between CLaMS-EI and CLaMS-M2 evolve after June.
Nützel et al. (2019) recently quantified water vapor transport from the ASM and NASM regions to the stratosphere by using CLaMS-EI simulations in a similar configuration. In our approach as well as that of Nützel et al. (2019), tracers are initialized in the upper troposphere above 350 K to reduce the impact of small-scale transport processes in the troposphere (e.g. convection) with uncertain representations in global reanalysis data. Our findings are also comparable to those of Vogel et al. (2019), who also used CLaMS-EI but with full transport from the boundary layer to the stratosphere included. In Table 2, we summarize
the methodologies (source, destination, time periods) and respective AOFs published by Vogel et al. (2019) and Nützel et al. (2019) in comparison with our results. The AOF from the NASM region to the LS-NH as reported by Nützel et al. (2019) is much smaller (4.4%) than 21% obtained in our study. The difference is due to different definitions of the LS-NH destination

| Source regions | | | | |
|---|---|---|---|---|
| Publication | ASM | NASM | Release time | Vert. range |
| Our study | 15−45° N, 30−120° E | 15−40° N, 160−60° W | Jul−Aug, 2010−13 | 350−360 K, 370−380K |
| Nützel et al. | 15−40° N, 20−130° E | 15−40° N, 170−60° W | Jul−Aug, 2010−13 | 360−380 K |
| Vogel et al. | India+China | – | May−Oct, 2007−08 | ABL (0−3 km) |

| Destination regions | | |
|---|---|---|
| Publication | LS-NH | TrP |
| Our study | 50−70° N, 340−380 K | 15° S−15° N, 480−550 K |
| Nützel et al. | 50−70° N, 400 K | 10° S−10° N, 450 K |
| Vogel et al. | 360 K, PV>5.5; 380 K, PV>7.2 | 30° S−30° N, 550 K |

| Comparison of transport | | | | |
|---|---|---|---|---|
| Publication | ASM→LS-NH | NASM→LS-NH | ASM→TrP | NASM→TrP |
| Our study | 18%, 22% | 10%, 21% | 7%, 4% | 2.5%, 2% |
| Nützel et al. | 22% | 4.4% | 12% | 5.2% |
| Vogel et al. | 18%, 16% | – | 6% | – |

**Table 2.** Comparison of the setups (source, destination, time periods) and the respective AOFs in Vogel et al. (2019), Nützel et al. (2019) and our study.

region (400 K level in Nützel et al. 2019; 340−380 K layer in our study), implying that transport from the NASM mainly affects the lowermost stratosphere, with relatively weak influences on the extratropical stratosphere above 400 K. Differences in AOF values quantifying transport to the TrP can also be explained by slightly different definitions of the TrP among these studies (see Table 2). However, while Nützel et al. (2019) restricted their analysis only to the monsoon pathway (by considering only transport from the 370−380 K layer), our study illustrates the importance of the tropical pathway in transporting air from the 350−360K layer in the monsoon regions to the TrP. On the other hand, Vogel et al. (2019) discussed the transport from the atmospheric boundary layer (ABL) over India/China to the TrP and did not distinguish between the tropical and monsoon pathways.

To more clearly separate the contributions of these two pathways to transport from the 350−360 K layer over the ASM to the TrP, we conduct an additional simulation in which we artificially suppress transport along the monsoon pathway by re-setting this tracer to zero in the 370−380 K layer over the ASM region through the full duration of the simulation. We find that more than 50% of the ASM tracers reaching the TrP (7% AOF) are transported via the tropical pathway (4% AOF). Moreover, the tropical pathway accounts for most of the transport from the ASM region to the LS-SH. The maximum AOF from the 350−360 K layer over the ASM to the LS-SH is 6%, of which more than 80% results from transport through the tropical pathway. Both tropical and monsoon pathways are schematically illustrated in Fig. 12. The strong heat source over Himalayas and Tibetan Plateau amplifies the northward shift of the ITCZ and forms the local monsoon Hadley circulation

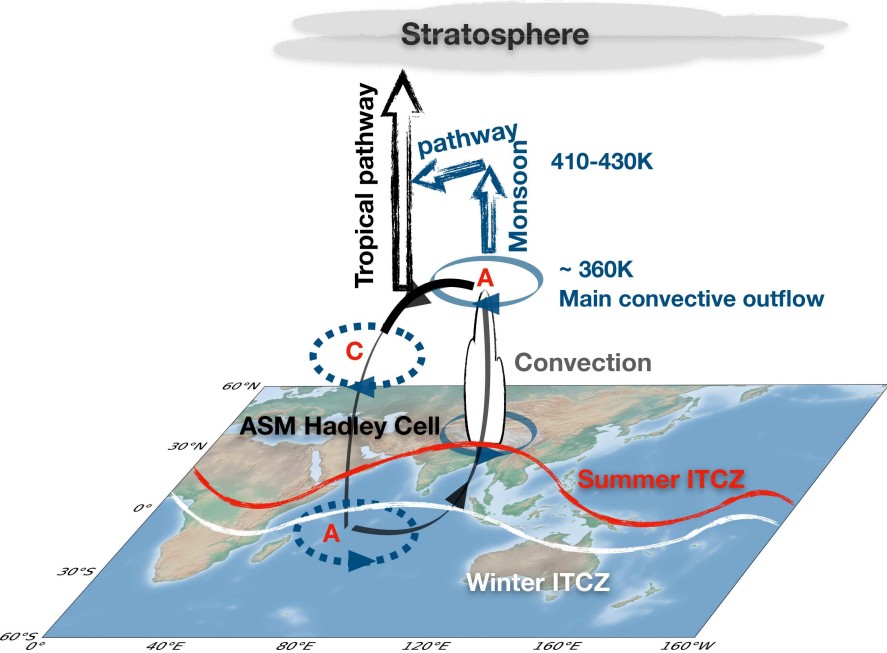

**Figure 12.** Schematic illustration of transport from ASM convective outflow along the main transport pathways to the stratosphere during boreal summer. The typical positions of the ITCZ and the monsoon Hadley cell are also shown.

(Molnar et al., 1993; Pillai and Mohankumar, 2008). The tracers from convective outflow within the ASM region ($\sim360$ K) can be transported to the tropics and even to the southern hemisphere via the upper-level cross-equatorial branch of the monsoon Hadley circulation, which is part of the tropical pathway in our study.

## 6 Conclusions

We have investigated the pathways, transit times and efficiencies of transport from the ASM and NASM regions to the stratosphere with tracers initialized at different vertical levels and advection driven by the ERA-Interim and MERRA-2 reanalyses. Both types of simulation show that transport from the upper troposphere ($350-360$ K) over ASM and NASM regions makes a larger contribution to the TrP relative to transport from just below the tropopause ($370-380$ K). Although part of the difference is related to the mass of air within each layer, which is larger for the $350-360$ K layer than for the $370-380$ K layer, there are also dynamical reasons related to the different strength of the anticyclone between these two levels. Thus, monsoon air mass fractions in the LS-SH and TrP during the following April−June are larger for tracers released at lower levels ($350-360$ K), reflecting weaker confinement by the anticyclone in the upper troposphere relative to just below the tropopause. By contrast, monsoon air fractions in the LS-NH region during the following April−June are larger for tracers released at higher lev-

els (370−380 K), reflecting the importance of wave-driven transport of tracers that reach the extratropical lower stratosphere quasi-isentropically at higher levels.

Peak values of monsoon tracers in the tropical stratosphere are in good agreement with the 'wet' phase of the water vapor tape recorder based on Aura MLS, which is consistent with the monsoon regions playing an influential role in the seasonal cycle of water vapor in the tropical lower stratosphere. Differences between simulations with tracers initialized at 350−360 K and simulations with tracers initialized at 370−380 K suggest the existence of two main transport pathways between the monsoon regions and the TrP. The first pathway (tropical pathway) involves the quasi-horizontal transport of monsoon air into the tropics, where that air then ascends into the stratosphere via tropical upwelling. The second pathway (monsoon pathway) involves the ascent of monsoon air within the monsoon anticyclone into the lower stratosphere, where it is then transported quasi-horizontally to the tropical lower stratosphere. Both pathways ultimately reach the TrP. However, the greater abundance of relatively young air along the tropical (first) pathway suggests that vertical transport is faster along this route than along the monsoon (second) pathway after summer.

CLaMS-EI and CLaMS-M2 results consistently indicate that the average final contribution of air mass from the ASM 350−360 K layer to the TrP (i.e. at the end of simulation period for each year, when all the tracer are reset to zero) is comparable to that of direct ascent from the inner tropics (∼5%). This contribution is approximately three times larger than the respective contribution from the NASM (∼1.5%). To eliminate the influence of the origin/source region size on our quantitative transport estimates, we also calculated the TE. The ASM region at 370−380 K shows the highest TE to the TrP (0.9‰) at the end of our simulation period. The respective TE from the NASM is similar to that from the tropics.

Our results further confirm the important role of the ASM anticyclone in troposphere-to-stratosphere transport. The maximum and final contributions as well the TE (Table 3) may help to explain, at least qualitatively, the stratospheric abundances of anthropogenic pollutants in relation to their lifetimes. Although the contribution from the NASM region to the TrP is less than the contributions from the ASM or from the tropics, it is still influential at about one third of the contribution from ASM. Moreover, the TE from the NASM to the TrP is as high as that of direct upwelling within the tropics during the boreal summer season.

*Data availability.* The MLS and ACE-FTS data are available at http://mirador.gsfc.nasa.gov and http://www.ace.uwaterloo.ca/data.php, respectively. The CLaMS model outputs may be obtained from the first author upon request.

*Author contributions.* XY carried out the ERA-Interim and MERRA-2 driven model simulations and the data analysis. PK, FP and AP contributed codes for the analysis. JW contributed codes to prepare the MERRA-2 reanalysis data. PK, FP and AP contributed to the design of the analysis. PK, FP, AP, JW, RM, MR provided helpful discussions and comments. XY wrote the paper with contributions from all co-authors.

|  |  | 350−360 K | | | 370−380 K | | |
| --- | --- | --- | --- | --- | --- | --- | --- |
| Destination |  | LS-NH | LS-SH | TrP | LS-NH | LS-SH | TrP |
| ASM | MC | 18 % | 6 % | 7 % | 22 % | 2 % | 4 % |
|  | ME | 2.5‰ | 0.4‰ | 0.7‰ | 9‰ | 0.35‰ | 1.45‰ |
|  | FC | 2.4 % | 3.2 % | 4.8 % | 1.2 % | 1.2 % | 3 % |
|  | FE | 0.5‰ | 0.2‰ | 0.55‰ | 0.6‰ | 0.2‰ | 0.89‰ |
|  | TT | 3 months | 6 months | 8.5 months | 2.5 months | 5.5 months | 8 months |
| NASM | MC | 10 % | 2.5 % | 2.5 % | 21 % | 1 % | 2 % |
|  | ME | 2.6‰ | 0.25‰ | 0.4‰ | 14‰ | 0.32‰ | 0.95‰ |
|  | FC | 1 % | 1.3 % | 2 % | 0.5 % | 0.6 % | 1.3 % |
|  | FE | 0.3‰ | 0.15‰ | 0.33‰ | 0.4‰ | 0.18‰ | 0.6‰ |
|  | TT | 2 months | 6 months | 9 months | 2 months | 5.5 months | 8 months |
| Tropics | MC | 17 % | 14 % | 10 % | 22 % | 14.5 % | 12 % |
|  | ME | 0.8‰ | 0.3‰ | 0.32‰ | 3.5‰ | 1.1‰ | 1.46‰ |
|  | FC | 1.2 % | 3.8 % | 5 % | 1.2 % | 3.5 % | 5 % |
|  | FE | 0.1‰ | 0.1‰ | 0.18‰ | 0.2‰ | 0.24‰ | 0.65‰ |
|  | TT | 2.5 months | 4.5 months | 8 months | 2.5 months | 4 months | 7.5 months |

**Table 3.** The maximum contribution (MC, in %), maximum efficiency (ME, in ‰), final (at the end of simulation period from each year) contribution (FC, in %), final efficiency (FE, in ‰) and transit time (TT, in month) from the three source regions (ASM, NASM, and Tropics) to the three destination regions (LS-NH, LS-SH, and TrP) based on the CLaMS-EI simulations.

*Competing interests.* The authors declare that they have no conflict of interest.

*Acknowledgements.* This research was partially supported by the National Key Research and development program of China (2018YFC1505703), the National Science Foundation of China (91837311, 41905040), and a joint DFG-NSFC research project with DFG project number 392169209 and NSFC project number 20171352419. We thank the European Centre for Medium-Range Weather Forecasts (ECMWF) and National Aeronautics and Space Admistration (NASA) for providing ERA-Interim and MERRA-2 meteorological reanalysis data for this study. We thank the MLS team for providing CO and $H_2O$ data, and the ACE-FTS team for providing HCN data. We thank Laura L. Pan for helpful discussions regarding the monsoon Hadley circulation. The authors would also like to thank the three anonymous reviewers for their very insightful comments.

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
