# Peer review of "The efficiency of transport into the stratosphere via the Asian and North American summer monsoon circulations"

_Atmospheric Chemistry and Physics, 2019_

## Referee Comment (RC1) · Anonymous Referee #3 · 17 Aug 2019

**General Comments**

Overall this is an important and well written paper that will be a serious contribution to the literature about the role of the monsoon transport in the UTLS region. I really liked the idea that there are two pathways and that the model statistics support those pathways

I think it would be helpful to summarize the efficiencies of the pathways and the differences in the models in a Table instead of text. Also Figure 11 should have the efficiency of the UT pathway to the base of the tropical pipe. The authors might also connect the efficiency of transport to the containment in the monsoons (see Pan et al, 2016, Trans-

port of chemical tracers from the boundary layer to stratosphere associated with the dynamics of the Asian summer monsoon, J. Geophys. Res. Atmos., 121, 14,159–14,174, doi:10.1002/ 2016JD025616.) who showed that the ASM is not as leaky as the NASM.

My major problem with this paper is I really don't understand the percentage argument used by the authors. Page 5 line 110 on the model set up confuses me. If I understand what the authors are doing is that they are starting up the model with some kind of uniform grid of parcels inside monsoon domain and the tropics. The model is running forward trajectories and then estimating the tracer ends up in each region. But as the system evolves, air from the SH will enter the tropics and air outside the monsoons will enter the monsoon region. The authors don't say how they account for this outside air in the estimates of the percentages after August 1. To be clear, I am not saying that the authors have done this wrong, but this paragraph gives me the impression that the CLaMS parcels are initiated over a limited domain. If this is true then it seems like the percentage estimates will be incorrect.

A second issue is that the authors initialize on July 1 of each year assuming that the monsoon develops about that time and then they stop tagging parcels after August 1. This seems like a limitation since the monsoon circulation can persist through early September. It seems to me some additional runs of the model would put to rest the sensitivity of their results to the limited tagging period.

Clearly Page 5 needs a lot of clarification. Since all of the rest of the paper is a function on how CLaMS was used here, I suggest the authors spend a little more time on the model set up and the assumptions behind it.

Minor comments: You don't need to tell us you used Python to make a figure.

---

## Referee Comment (RC2) · Anonymous Referee #1 · 26 Aug 2019

**1   General comments**

The paper is an interesting and important contribution for assessment of the monsoon influence on the global stratosphere and should be published in ACP. New against earlier studies (e.g. Ploeger et al., 2017) is the analysis of NASM effects. However, the paper would gain a lot if simulations on the sensitivity of results on the positions of the boundaries of the monsoon boxes are included, especially the southern edge of ASM, which appears to be rather close to the equator, and the eastern edge of NASM (too far east). The position of the southern edge of ASM (e.g. 20N instead of 15N) might be critical for transport to the tropical pipe and to the southern hemisphere

as indicated in Yu et al. (2017). This should be included in Figures 8 and 9. Also some sentences should address the differences to the method where the monsoon air masses are separated using PV instead of a rectangular box and if results are different to Garny and Randel (2016).

**2 Specific comments**

In Figure 1 it would be more useful to show CO for 350 or 360K. At 340K the reader is distracted by the large effects of biomass burning in the tropics.

Line 110: For clarity say 'tracer is reset to zero every...'. This should also go into the caption of Fig. 5.

Line 129ff: Please define more clearly on what the percentage mass fraction is based. Are the masses in the monsoon boxes during July and August 100%?

Line 199: What is annual mean here? Is there each year different or is the mean of the 4-year time series meant? It might be sufficient just to show the MLS-data.

**3 Technical corrections**

Line 17: Remove 'in contrast'.

Figure 1a: The rather arbitrary spacing of contours and colors should be more systematic (e.g. linear or a function mentioned in the caption) and fit to each other. Here also less colors are more.

Figure 3 and 4: Enhance sizes for legends and labels.

Figure 5 and 6: Avoid smoothing artefact when tracer is reset, include 'zonal' in caption.

Check 'Competing interests'.

References: Correct Latex errors and remove second link for several entries.

---

## Referee Comment (RC3) · Anonymous Referee #2 · 9 Sep 2019

This manuscript uses CLaMS simulations driven by both ERA-Interim and MERRA-2 to quantify the relative contributions and transport efficiencies from different atmospheric layers in the Asian and North American summer monsoon regions to the stratosphere. Artificial tracers elucidate the main transport pathways from the two monsoon source regions and the Tropics to three destination regions: the tropical pipe and the extratropical lower stratosphere in both hemispheres. The manuscript is very well written and well prepared, and the analysis is sound. The results will be of interest to the broad ACP readership. However, there are a few issues (mostly minor points of clarification) that I would like to see addressed before the manuscript is published.

[Figure]

General comments:

(1) Somewhere in this manuscript (probably in multiple places, including the introduction, main discussion of results, and conclusions), the authors need to relate their findings to those of Vogel et al. [2019], who also used CLaMS artificial tracers to study transport of pollutants and the pathways by which air masses enter the tropical pipe via the ASM circulation. While that paper is cited as part of a list of references in the first paragraph of the Introduction, it is never mentioned again. Although the details of the two analysis approaches differ, I feel that some discussion of how the results from the current manuscript fit in with the concept of the "upward spiraling range" introduced by Vogel et al. [2019], as well as comparisons with the transit times and fraction of ASM air masses in the TrP that they calculated, is warranted here.

(2) The authors tend to cite one or two papers for well-established points, without adding "e.g." at the front of their short list. Obviously not all relevant papers can or even should be cited, but I feel that overlooking the literature to this extent does a disservice to both the authors (because it erroneously reflects poorly on the depth of their knowledge of the field) and the previous studies, so I encourage them to make a greater effort in referencing prior work. Some places where the lack of references particularly bothered me are called out in the specific comments below.

Specific substantive comments and questions:

L30: Liang et al. [2004], while not an inappropriate reference for the point that ASM air can be transported to distant locations, should certainly not be the ONLY paper cited for this point – in fact, there are probably at least a dozen papers that could be added here.

L47-64: Although it is true that the NASM has received much less scientific attention than the ASM, it has not been neglected quite to the extent implied by these two paragraphs. I think that the authors should do a more thorough job of summarizing previous work on the influence of the NASM on UTLS composition. Given that water vapor is of

particular interest in this manuscript, Anderson et al. [Science, 2012], Schwartz et al. [GRL, 2013], and Randel et al. [JGRA, 2015] should be mentioned. More generally, other references to consider including are: Q. Li et al. [JGR, 2005], Cooper et al. [JGR, 2006, 2007], Barth et al. [ACP, 2012], etc.

Figure 1b: The 340 K surface seems too low to be appropriate for the map of MLS CO, which is not recommended for scientific use at pressures greater than 215 hPa. In fact, if proper data quality screening were applied, I would expect to see much of the tropics blanked out in a map of CO at this level. Thus some of the "hot spots" in this panel may be suspect. Since the specific level shown in this figure is not critical, I suggest 350 K instead.

L110-111: Some of the choices made here, while no doubt perfectly legitimate, should be explained. For instance, why are the simulations run over the 3-year period from 2010 to 2013? That is, why three years (and not two or ten), and why those particular years? Why are the tracers initialized over the interval 1 July through 31 August, when the anticyclone spins up by the beginning of June (if not earlier) and persists through September in most years? I am not suggesting that the analysis based on these choices is flawed, merely that they need to be better justified.

L131: Similarly, why is the April-to-June period used for Figures 3 and 4? I assume that June was chosen as the end of the interval because the tracers are re-initialized at the start of the next ASM season in July. But why include results starting in April (and not March or May)?

L155-161: References should be given for all three of the effects listed in L157-158. I see why these factors would lead to higher column amounts in the SH at 350-360 K than in the SH at 370-380 K. But it is less clear to me why they would lead to higher column amounts in the SH at 350-360 K than in the NH at that level.

L203-204: Other references would be appropriate here as well, including Bannister et al. [QJRMS, 2004], James et al. [GRL, 2008], and Dethof et al. [1999] (already cited

elsewhere in the manuscript), etc.

L215: Again, a brief explanation of why the 400 K level is selected to be shown in Figure 6 might be good. Also, it might be helpful to add a horizontal line (maybe dashed or in grey) at this level in Figure 5, to orient the reader for the following plot.

Figure 7: Have the results in this figure been aggregated over the 2010-2013 period? How was the particular interval shown (November-May) chosen? It would be good to define what is meant by "young" in the figure caption as well as the main text. As stated in L239-240, the tropical pathway is more common for tracers released at 350-360 K, but it does not appear to be entirely absent for the 370-380 K tracers in Figure 7c. There are hints of a "fork" in the ASM tracer distribution between ∼3.7-4.0 ppmv and ∼2% (in which case the cyan arrow may be slightly misplaced). There may even be a faint hint of similar structure for the NASM tracer (Figure 7d), but the cyan arrow, useful though it is, obscures it.

L262-263: Is this time difference consistent with known upwelling rates? (A reference would be good.)

Figure 8: I understand that scaling the standard deviations improves the legibility of the plot, but multiplying by 0.2 seems like a fairly drastic step that produces a misleading impression of the degree of variability. How can such a substantial reduction in the scatter in this plot be justified? If the full envelopes were presented, results for the various destination regions would likely overlap significantly. As it is, I fear that the figure instills more confidence in the separability of the regions than is really warranted.

L314-318: To my eye, the TE into the LS-SH is never dominated by ASM or NASM sources for tracers released at 370-380 K – after February, the curves for all three sources lay nearly on top of one another. Moreover, for the ASM tracers transport from the 350-360 K layer dominates over that from the Tropics starting in December, not January. Finally, the TE from the ASM is nearly 50% larger than that from the NASM, so perhaps "slightly" should be deleted in L318.

L340-341: It would be appropriate to include here some references for the effects of Rossby wave breaking and eddy shedding on mixing monsoon air into the extratropics.

L404-411: I was confused the first couple of times that I read this paragraph, because I expected the results cited here to have been shown in Figure 11 – it is the last figure in the paper and freshest in readers' minds when they arrive at the Conclusions. I hadn't understood what was meant by "ultimate" in L404 (in fact, I don't think that the usage of that word conveys quite what the authors intend), and so it took me several minutes to realize that the numbers being quoted here for the most part refer to the end of the simulation period in Figures 8 or 9 and thus do not match the values in Figure 11. I concede that I obviously was not reading these sentences carefully enough, but I'm guessing that many readers may do the same and also may fail to note that Figure 11 shows the "maximum" contributions/efficiencies. That information is noted in the figure caption, but it is not stated when this figure is introduced in L358, which instead describes it as showing "overall contributions, efficiencies, and transit times". In addition, stating values such as 0.9 for the TE in L410 without specifying that this value refers to the end of the simulation compounds the confusion, as does stating a range for the TE from the Tropics to the TrP. In my mind this entire discussion needs to be clarified, with a bit more hand-holding to help the reader follow the details. However, this brings up a philosophical question about whether showing the maximum contributions/efficiencies is really the best approach for Figure 11. Moreover, while reading this paragraph I also wondered why a similar panel for the Tropics was not included in that figure.

Minor points of clarification, wording suggestions, and grammar / typo corrections:

L30: influences –> influence

Figure 1 caption: I questioned the need for the seemingly unimportant detail about the map being produced by python in my initial access review, and I still don't see why this information is useful to the reader. A similar comment applies to Figure. 10.

L57: "Meanwhile" seems like an odd choice of word here

L93: add a comma after "anticyclone"

L116: TrP has already been defined (L41)

Figure 3 and caption. Although it is stated in the main text, it would be good to add "in July and August" somewhere in the caption, perhaps after "initialized" or before "in CLaMS-E1". Also, some odd glitches are apparent in the dashed line in this figure, especially in panel 3b at about (45N, 10m).

L155: The interhemispheric difference is fairly small, especially for the total column, so I suggest adding "slightly" in front of "larger"

L156: since this sentence is about the SH, just to be really clear, add "boreal" in front of "monsoon"

L158-159: portion . . . enters . . . and is (not "enter" and "are")

L173: add a comma after "simulations"

L177: "not shown" – is this point not shown by comparison of Figures 3 and 4?

Figure 5 caption: I think it would be helpful to add "over the July 2010 to April 2014 period" after "sections".

L210: . . . tracers is slightly lower –> . . . tracers is slightly weaker

L221: "spread out" might be better than "widespread"

L239: ASM (NASM) –> ASM (NASM) region

L255: show –> shows

L264-266: it would draw the contrast (and flow) better to move "after three months" to right after "However," at the beginning of the sentence.

Figure 8 caption: I think it might work better to say ". . . simulations of air mass fractions (in %) in three source regions"

L268: that –> those

L271: it might be good to add "throughout the year" at the end of this sentence

L279-280: it might be good to add "As for the ASM," at the beginning of this sentence

L289: it might be good to add "Much" in front of "more air"

L291: delete "and"

L297: delete "up to and"

L321: To me, "after March" means "starting in April", but in fact the NASM TE exceeds the tropical TE in the TrP region at the beginning of March for the 350-360 K tracers. Thus "after March" should be "by March". Similarly, "after April" should be "by April". In addition, there is a typo at the end of this line: 380 KIn –> 380 K. In

L325-326: that –> those. Also, the CLaMS-M2 figure is omitted so I cannot judge myself, but I assume that a similar issue to the point raised above exists for "after December . . . or January".

L337-339: these two sentences are somewhat redundant and could be combined for efficiency (and to eliminate the slightly awkward construction ". . . Fig. 10. Figure 10 . . ."). Also, when were the results for 24 August 2012 shown in this figure initialized?

L342-343: replace the second instance of "CLaMS-EI and CLaMS-M2" in this line with "the two simulations"

L366: it would be good to remind readers of these pathways by adding "(monsoon and tropical)" after "pathways"

L371: It is very confusing to start this sentence with "As for the NASM". This kind of construction is often used to set up a discussion of similarity, but the previous sentence is also talking about the NASM, so that doesn't make sense. You may have meant "As is the case for the ASM", in which case there is a typo ("NASM" should be "ASM").

That's what I assumed the first time I read this sentence, so I suggested making that change in my access review. Since the phrase remains in this version, I am guessing that was not your intention, and thus it is probably best to simply delete this phrase.

L375: maybe add "(not shown)" again at the end of the sentence

L386: I feel that the Conclusions section starts too abruptly – it needs some sort of introductory sentence to set the stage and sum up what was done in the paper. On the other hand, such a sentence is not really needed at the beginning of the Discussion section. Thus I suggest moving the first sentence in that section ("We have investigated . . .", L330-331) here.

L389-390: "vertical differences" is awkward. I suggest instead "differences in the dynamical situation with altitude"

References: the doi's for many of the references are repeated.

---

## Author Comment (AC1) · 20 Nov 2019

Many thanks to the reviewer for the comments, and they have helped to improve the clarity of the manuscript. In the following, we address all the points raised in the review (denoted by italic letters). Text changes in the manuscript are highlighted in red or blue.

**General comments**

*Overall this is an important and well written paper that will be a serious contribution to the literature about the role of the monsoon transport in the UTLS region. I really liked the idea that there are two pathways and that the model statistics support those pathways.*

1. *I think it would be helpful to summarize the efficiencies of the pathways and the differences in the models in a Table instead of text. Also Figure 11 should have the efficiency of the UT pathway to the base of the tropical pipe. The authors might also connect the efficiency of transport to the containment in the monsoons (see Pan et al, 2016, Transport of chemical tracers from the boundary layer to stratosphere associated with the dynamics of the Asian summer monsoon, J. Geophys. Res. Atmos., 121, 14,159−14,174, doi:10.1002/2016JD025616.) who showed that the ASM is not as leaky as the NASM.*

   A. We summarize the contribution and efficiency of transport and transit time from the source regions to the destination regions in the end of the manuscript. The numbers in red color in Fig.11 represent the sum of transport along monsoon and tropical pathways from $350−360$ K over monsoon regions to the tropical pipe. The separation of the contribution of transport along different pathways to the tropical pipe is included in the discussion section with more than 50% of contributions from tropical pathway. Regarding the last point, we didn't find the work about NASM from Pan et al., 2016. We include two more subplots from MLS CO in the revised manuscript. Indeed, the ASM is not as leaky as the NASM. We also connected this point to the transport to the LS-NH in Section 4.

2. *My major problem with this paper is I really don't understand the percentage argumentn used by the authors. Page 5 line 110 on the model set up confuses me. If I understand what the authors are doing is that they are starting up the model with some kind of uniform grid of parcels inside monsoon domain and the tropics. The model is running forward trajectories and then estimating the tracer ends up in each region. But as the system evolves, air from the SH will enter the tropics and air outside the monsoons will enter the monsoon region. The authors don't say how they account for this outside air in the estimates of the percentages after August 1. To be clear, I am not saying that the authors have done this wrong, but this paragraph gives me the impression that the CLaMS parcels are initiated over a limited domain. If this is true then it seems like the percentage estimates will be incorrect.*

   A. The text was not so clear about the setup of the model. The tracer set-up is the same as in Orbe et al. (2015), Ploeger et al. (2017). We set the artificial tracer mixing ratio as 1 inside the monsoon regions and the tropics during July and August. The model simulations are driven by horizontal winds and diabatic heating from reanalysis. We estimate the abundance of source tracer at any location in the atmosphere. The percentage in our study represents the monsoon/tropical tracer mixing ratio at any location, and it equals the fraction of air which left the corresponding source layer in the ASM, NASM or tropical domain during the previous monsoon season. The transport from different source regions is simulated independently in

our study. Therefore, there is no interactive influence among the transport from the three source regions. Based on the comment, we rephrase the text in the manuscript to make it more clear.

3. ***A second issue is that the authors initialize on July 1 of each year assuming that the monsoon develops about that time and then they stop tagging parcels after August 1. This seems like a limitation since the monsoon circulation can persist through early September. It seems to me some additional runs of the model would put to rest the sensitivity of their results to the limited tagging period.***

A. We run the simulation for the same period as Ploeger et al. (2017) to have the direct comparison from different definitions of the domain boundary (simplified box and PV-barrier). As suggested by the Reviewer, we now added a sensitivity simulation where we set the artificial tracer as 1 from 15 June to 15 September and run the simulation. Figure RL 1 shows the average air mass fraction from ASM and NASM regions

[Figure]

Figure RL 1: Climatological (2010−2013) zonal mean air mass fraction from the ASM (a and c) and the NASM (b and d) initialized at 350−360 K (upper) and 370−380 K (lower) in CLaMS-EI (color shading) during the following April−June. HCN from ACE-FTS observations (black contours) is also shown for context. Regions with HCN volume mixing ratios greater than 215 pptv are hatched. Blue lines mark the lapse rate tropopause. (Note the logarithmic color scale.)

initialized in the 350−360 K and 370−380 K layers during 15 June−15 September of 2010−2013 in the CLaMS-EI simulations during the following April−June. We can see a lot of similarities between Figure RL 1 and Fig.3 in the manuscript. In particular the transport patterns are not depending on the exact length of the initialization period. More monsoon tracer is transported into the stratosphere from the 350−360 K layer than from the 370−380 K layer for both the ASM and NASM regions. The abundance of monsoon air initialized at 350−360 K is higher in the SH stratosphere than in the NH, while monsoon air initialized at 370−380 K is more likely to remain in the NH. The exact value of monsoon air mass fraction in the stratosphere is higher for the the longer initialization period.

As Fig.8 in the manuscript, we also quantify the transport from source regions to the destination regions

[Figure]

Figure RL 2: Climatological time series based on the CLaMS-EI simulations of source regions: (a) ASM and (b) NASM air mass fractions (in %) and diagnosed in three destination regions (see Fig.2 in the manuscript): tropical pipe (TrP, solid line), extratropical lower stratosphere in the NH (LS−NH, dashed line) and in the SH (LS−SH, dotted line). Shading shows the mean standard deviation in the zonal average (multiplied by 0.2 for better visibility). Red and blue lines respectively represent the tracers released in the 350−360 K and 370−380 K layer.

(Figure RL 2). The abundances of monsoon air in the three destination regions are larger than those from Fig.8 in the manuscript because of the longer simulation period. However, the transport patterns are quite similar. July and August is at the mature phase of the monsoon circulations. Using this period can transport less in-mixing of air from the adjacent regions of the monsoon regions. The simulation results from July and August include most of the features from longer period. The same range of simplified box domain of monsoon regions used in June and September may overestimate the contribution of transport from monsoon regions. The contributions of transport from ASM region to the TrP and LS-NH quantified in our work based on July-August period show similar values with the previous study covering the time period of May−October Vogel et al., 2019. Hence, we simulate the monsoon transport during July-August instead of June-September. We added a remark on this sensitivity in the discussion section of revised manuscript (P22).

4. *Clearly Page 5 needs a lot of clarification. Since all of the rest of the paper is a function on how CLaMS was used here, I suggest the authors spend a little more time on the model set up and the assumptions behind it.*

A. Thanks for pointing this out. More details about model setup and assumptions are included in the revised version of the draft.

5. ***Minor comments: You don't need to tell us you used Python to make a figure.***

A. The information about Python is removed.

---

## Author Comment (AC2) · 20 Nov 2019

Many thanks to the reviewer for the comments, and they have helped to improve the clarity of the manuscript. In the following, we address all the points raised in the review (denoted by italic letters). Text changes in the manuscript are highlighted in red or blue.

**1. General comments**

*The paper is an interesting and important contribution for assessment of the monsoon influence on the global stratosphere and should be published in ACP. New against earlier studies (e.g. Ploeger et al., 2017) is the analysis of NASM effects. However, the paper would gain a lot if simulations on the sensitivity of results on the positions of the boundaries of the monsoon boxes are included, especially the southern edge of ASM, which appears to be rather close to the equator, and the eastern edge of NASM (too far east). The position of the southern edge of ASM (e.g. 20N instead of 15N) might be critical for transport to the tropical pipe and to the southern hemisphere as indicated in Yu et al. (2017). This should be included in Figures 8 and 9. Also some sentences should address the differences to the method where the monsoon air masses are separated using PV instead of a rectangular box and if results are different to Garny and Randel (2016).*

A. Thanks for this important remark. We now added a sensitivity simulation with the ASM source region defined as [20° N, 45° N, 30° E, 120° E] and the NASM source region defined as [15° N, 40° N, 120° W,

[Figure]

Figure RL 1: Climatological (2010−2013) zonal mean air mass fraction from the ASM (a and c) and the NASM (b and d) initialized at 350−360 K (upper) and 370−380 K (lower) in CLaMS-EI (color shading) during the following April−June. HCN from ACE-FTS observations (black contours) is also shown for context. Regions with HCN volume mixing ratios greater than 215 pptv are hatched. Blue lines mark the lapse rate tropopause. (Note the logarithmic color scale.)

60° W]. Figure RL 1 shows the average air mass fraction from ASM and NASM regions initialized in the 350−360 K and 370−380 K layers in July and August of 2010−2013 in the CLaMS-EI simulations during the following April−June. We can see a lot of similarities between Figure RL 1 and Fig.3 in the manuscript. In particular the transport patterns are not depending on the exact boundary of the source region. Here, more monsoon tracer is transported into the stratosphere from the 350−360 K layer than from the 370−380 K layer for both the ASM and NASM regions. The abundance of monsoon air initialized at 350−360 K is higher in the SH stratosphere than in the NH, while monsoon air initialized at 370−380 K is more likely to remain in the NH. The monsoon air mass in the stratosphere is lower than it from Fig.3 in the manuscript mainly because of the smaller domain used here.

To quantify the transport from source regions with different domain to the destination regions, we do the

[Figure]

Figure RL 2: Climatological time series based on the CLaMS-EI simulations of source regions: (a) ASM and (b) NASM air mass fractions (in %) and diagnosed in three destination regions (see Fig.2 in the manuscript): tropical pipe (TrP, solid line), extratropical lower stratosphere in the NH (LS-NH, dashed line) and in the SH (LS-SH, dotted line). Shading shows the mean standard deviation in the zonal average (multiplied by 0.2 for better visibility). Red and blue lines respectively represent the tracers released in the 350−360 K and 370−380 K layer.

same as Fig.8 in the manuscript. Figure RL 2 shows CLaMS-EI time series of ASM and NASM tracer in the three destination regions (TrP, LS-NH and LS-SH). The abundances of ASM air in the three destination regions are slightly lower than those from Fig.8 in the manuscript as we expect because of the smaller domain. While the contributions from NASM to the LS-NH are much smaller compared to the results from Fig.8 in the manuscript because the domain used in the manuscript is almost twice larger than the

domain here, we can see better comparison results from the transport efficiency (TE) below. However, the transport patterns are quite similar including the daily structures. Hence, the conclusions of our study are qualitatively robust to reasonable definition of the monsoon boundary.

Figure RL 3 compares the TE from the ASM and NASM source regions each divided into two layers

[Figure]

Figure RL 3: Climatological time series of transport efficiency (TE) as derived from the CLaMS-EI simulations and calculated for transport from the ASM (solid line)and NASM (dashed line) source regions to the (a) LS-NH , (b) LS-SH and (c) TrP destination regions. TE is defined by normalizing the average monsoon or tropical air mass over the destination region by the mass of the source region. Different colors of the lines represent tracers released at different levels.

(350−360 K and 370−380 K) to the three destination regions. Comparing the TE here and the TE from Fig.9 in the manuscript, we can see that the TEs from ASM/NASM to the three destination regions are slightly larger than those from Fig.9, especially for the NASM source region. The differences are small, we included a remark of this result in the discussion section (P22) of the manuscript instead of changing the figures.

Regarding the last point, the comparison results between our study and the results from Ploeger et al.(2017)

and Garny and Randel (2016) are included in the revised version of the manuscript. There is good agreement with the work from Ploeger et al.(2017), we also include the difference in the manuscript. We can not compare the contributions of transport to different destinations from our study with the results from Garny and Randel (2016) because we use different method, and the meaning of the numbers in their work are also different from ours. The main comparison is related to the transport pathways to different destination regions from 360 K and 380 K. The insensitivity of the results to the source domain boundary in our study is in agreement with the findings shown by Garny and Randel (2016), where they found that the differences are small regarding a change in the source region definition from PV-contour to box.

**2. **Specific comments**

(1) *In Figure 1 it would be more useful to show CO for 350 or 360K. At 340K the reader is distracted by the large effects of biomass burning in the tropics.*

A. The CO at 350 K is shown in Fig.1, the text is changed correspondingly.

(2) *Line 110: For clarity say 'tracer is reset to zero every...'. This should also go into the caption of Fig. 5.*

A. This is included in the caption of Fig.5 and Fig.6.

(3) *Line 129ff: Please define more clearly on what the percentage mass fraction is based. Are the masses in the monsoon boxes during July and August 100%?*

A. The percentage mass fraction is based on the source air mass concentration (mixing ratio) in the stratosphere. Yes, the air masses in the monsoon boxes during July and August are 100%. The text is changed as well.

(4) *Line 199: What is annual mean here? Is there each year different or is the mean of the 4-year time series meant? It might be sufficient just to show the MLS-data.*

A. The annual mean used here is different from each year. Figure RL 4 and Figure RL 5 respectively show the vertical and horizontal distribution of monsoon tracers together with the original MLS water vapor. We can see that the connection between monsoon tracers and MLS water vapor looks clear if we removed the annual mean.

[Figure]

Figure RL 4: Potential temperature−time sections of mean monsoon air fraction (color shading) between 15° S and 15° N released in the 350−360 K (upper; a and b) and 370−380 K (lower; c and d) layers within the ASM (left; a and c) and NASM (right; b and d) based on CLaMS-EI simulations. Tape-recorder signals calculated from Aura MLS water vapor (in ppmv) are shown as black contours for context. Note that the tracer is set to zero everywhere on 1 July of each year.

[Figure]

Figure RL 5: Zonal-mean tracer concentrations (color shading) for the ASM (left; a and c) and NASM (right; b and d) tracers initialized at 350−360 K (upper; a and b) and 370−380 K (lower; c and d) as functions of time and latitude on the 400 K isentropic surface based on CLaMS-EI. MLS water vapor retrievals (in ppmv) are shown as black contours for context. Note that the tracer is set to zero everywhere on 1 July of each year.

**3. Technical corrections**

(1) *Line 17: Remove 'in contrast'.*

A. It is removed in the manuscript.

(2) *Figure 1a: The rather arbitrary spacing of contours and colors should be more systematic (e.g. linear or a function mentioned in the caption) and fit to each other. Here also less colors are more.*

A. The colorbar and contour space are changed in Fig.1.

(3) *Figure 3 and 4: Enhance sizes for legends and labels.*

A. The size of the legends and labels are enhanced in Fig.3 and Fig.4.

(4) *Figure 5 and 6: Avoid smoothing artefact when tracer is reset, include 'zonal' in caption.*

A. The data is not smoothed. 'zonal' is included in the caption of Fig.5 and Fig.6.

(5) *Check 'Competing interests'.*

A. 'Competing interests' is included in the manuscript.

(6) *References: Correct Latex errors and remove second link for several entries.*

A. The errors about the references are corrected, and the second link of each citation is removed.

---

## Author Comment (AC3) · 20 Nov 2019

Many thanks to the reviewer for the comments, and they have helped to improve the clarity of the manuscript. In the following, we address all the points raised in the review (denoted by italic letters). Text changes in the manuscript are highlighted in red or blue.

**1. General comments**

(1) *Somewhere in this manuscript (probably in multiple places, including the introduction, main discussion of results, and conclusions), the authors need to relate their findings to those of Vogel et al. [2019], who also used CLaMS artificial tracers to study transport of pollutants and the pathways by which air masses enter the tropical pipe via the ASM circulation. While that paper is cited as part of a list of references in the first paragraph of the Introduction, it is never mentioned again. Although the details of the two analysis approaches differ, I feel that some discussion of how the results from the current manuscript fit in with the concept of the "upward spiraling range" introduced by Vogel et al. [2019], as well as comparisons with the transit times and fraction of ASM air masses in the TrP that they calculated, is warranted here.*

A. Thanks for making us aware of this missing discussion. A more detailed comparison to the results of Vogel et al. [2019] is now included in the discussion section. The changes include comparable abundance of ASM air in the LS-NH from Vogel et al. [2019] (18% and 16% from 360 K and 380 K) and from our work (22% and 18%). The ASM air in the tropical pipe from Vogel et al. [2019] is 6%, which is also similar to our results (4%-6%). The transit time from ASM to the tropical pipe is about 8-9 months, which is consistent with the result from Vogel et al. [2019] with the transit time within ∼1 year. The monsoon pathway from ASM at 350-360 K to the tropical pipe is inside the "upward spiraling range" introduced by Vogel et al. [2019], the tracers may transport upward in spiral rising way. However, the tracer setting between the work from Vogel et al. [2019] and our study is different, e.g. source regions, vertical range (boundary layer versus upper troposphere), and initialization periods (May-Oct versus Jul-Aug). Therefore the differences of the exact numbers are expected.

(2) *The authors tend to cite one or two papers for well-established points, without adding "e.g." at the front of their short list. Obviously not all relevant papers can or even should be cited, but I feel that overlooking the literature to this extent does a disservice to both the authors (because it erroneously reflects poorly on the depth of their knowledge of the field) and the previous studies, so I encourage them to make a greater effort in referencing prior work. Some places where the lack of references particularly bothered me are called out in the specific comments below.*

A. We change the text about the reference. We include more references or put "e.g." in front of some citations.

**2. Specific substantive comments and questions:**

(1) *L30: Liang et al. [2004], while not an inappropriate reference for the point that ASM air can be transported to distant locations, should certainly not be the ONLY paper cited for this point − in fact, there are probably at least a dozen papers that could be added here.*

A. We include more references in the manuscript.

(2) ***L47-64: Although it is true that the NASM has received much less scientific attention than the ASM, it has not been neglected quite to the extent implied by these two paragraphs. I think that the authors should do a more thorough job of summarizing previous work on the influence of the NASM on UTLS composition. Given that water vapor is of particular interest in this manuscript, Anderson et al. [Science, 2012], Schwartz et al. [GRL, 2013], and Randel et al. [JGRA, 2015] should be mentioned. More generally, other references to consider including are: Q. Li et al. [JGR, 2005], Cooper et al. [JGR, 2006, 2007], Barth et al. [ACP, 2012], etc.***

A. More references are included in the manuscript, and the text is rephrased.

(3) ***Figure 1b: The 340 K surface seems too low to be appropriate for the map of MLS CO, which is not recommended for scientific use at pressures greater than 215 hPa. In fact, if proper data quality screening were applied, I would expect to see much of the tropics blanked out in a map of CO at this level. Thus some of the "hot spots" in this panel may be suspect. Since the specific level shown in this figure is not critical, I suggest 350 K instead.***

A. We agree the comment from the reviewer. We take the CO at 350 K in the revised version of the manuscript.

(4) ***L110-111: Some of the choices made here, while no doubt perfectly legitimate, should be explained. For instance, why are the simulations run over the 3-year period from 2010 to 2013? That is, why three years (and not two or ten), and why those particular years? Why are the tracers initialized over the interval 1 July through 31 August, when the anticyclone spins up by the beginning of June (if not earlier) and persists through September in most years? I am not suggesting that the analysis based on these choices is flawed, merely that they need to be better justified.***

A. We run the simulation at the same period as Ploeger et al. (2017) to have the direct comparison from different definitions of the domain boundary (simplified box and PV-barrier). We now added a sensitivity simulation where we set the artificial as 1 from 15 June to 15 September and run the simulation. Figure RL 1 shows the average air mass fraction from ASM and NASM regions initialized in the $350-360$ K and $370-380$ K layers during 15 June$-$15 September of $2010-2013$ in the CLaMS-EI simulations during the following April$-$June. We can see a lot of similarities between Figure RL 1 and Fig.3 in the manuscript. Here, more monsoon tracer is transported into the stratosphere from the $350-360$ K layer than from the $370-380$ K layer for both the ASM and NASM regions. The abundance of monsoon air initialized at $350-360$ K is higher in the SH stratosphere than in the NH, while monsoon air initialized at $370-380$ K is more likely to remain in the NH. The monsoon air mass fraction in the stratosphere is higher than it from Fig.3 in the manuscript due to the longer simulation period here.

As Fig.8 in the manuscript, we also quantify the transport from source regions to the destination regions (Figure RL 2). The abundances of monsoon air in the three destination regions are larger than those from Fig.8 in the manuscript because of the longer simulation period. However, the transport patterns are quite similar. July and August is at the mature phase of the monsoon circulations. Using this period can transport less in-mixing of air from the adjacent regions of the monsoon regions. The simulation results from July and August include most of the features from longer period. The same range of simplified box domain of monsoon regions used in June and September may overestimate the contribution of transport from monsoon regions. The contributions of transport from ASM region to the TrP and LS-NH quantified in our work based on July-August period show similar values with

[Figure]

Figure RL 1: Climatological (2010−2013) zonal mean air mass fraction from the ASM (a and c) and the NASM (b and d) initialized at 350−360 K (upper) and 370−380 K (lower) in CLaMS-EI (color shading) during the following April−June. HCN from ACE-FTS observations (black contours) is also shown for context. Regions with HCN volume mixing ratios greater than 215 pptv are hatched. Blue lines mark the lapse rate tropopause. (Note the logarithmic color scale.)

the previous study covering the time period of May−October Vogel et al., 2019. Hence, we simulate the monsoon transport during July-August instead of June-September. We change the text and mainly include this point at the "Discussion" section of the manuscript.

(5) *L110-111: L131: Similarly, why is the April-to-June period used for Figures 3 and 4? I assume that June was chosen as the end of the interval because the tracers are re-initialized at the start of the next ASM season in July. But why include results starting in April (and not March or May)?*

A. We would like to show the evolution of the monsoon tracers from July to June in the next year. The evolution process shows a lot of similarities with the results in Ploeger et al. (2017). To avoid the redundancy, we just show the result from April-June because it includes the transport to the tropical pipe which is import to the global stratosphere. We include the explanation in the revised version of the manuscript.

(6) *L155-161: References should be given for all three of the effects listed in L157-158. I see why these factors would lead to higher column amounts in the SH at 350-360 K than in the SH at 370-380 K. But it is less clear to me why they would lead to higher column amounts in the SH at 350-360 K than in the NH at that level.*

A. The references are added in the manuscript. The higher column of monsoon air in the SH from 350-360 K is more connected to the seasonality of the Brewer-Dobson circulation. The monsoon tracers are compressed to the lower altitude with higher density in the SH compared to the monsoon tracers in the NH, this leads to the higher columns of the monsoon tracer in the NH at the end.

[Figure]

Figure RL 2: Climatological time series based on the CLaMS-EI simulations of source regions: (a) ASM and (b) NASM air mass fractions (in %) and diagnosed in three destination regions (see Fig.2 in the manuscript): tropical pipe (TrP, solid line), extratropical lower stratosphere in the NH (LS-NH, dashed line) and in the SH (LS-SH, dotted line). Shading shows the mean standard deviation in the zonal average (multiplied by 0.2 for better visibility). Red and blue lines respectively represent the tracers released in the $350-360\,$K and $370-380\,$K layer.

(7) *L203-204: Other references would be appropriate here as well, including Bannister et al. [QJRMS, 2004], James et al. [GRL, 2008], and Dethof et al. [1999] (already cited elsewhere in the manuscript), etc.*

A. The references are added.

(8) *L215: Again, a brief explanation of why the 400 K level is selected to be shown in Figure 6 might be good. Also, it might be helpful to add a horizontal line (maybe dashed or in grey) at this level in Figure 5, to orient the reader for the following plot.*

A. We choose 400 K to investigate the monsoon influence on the lowermost stratosphere. 380 K level is widely used to study the lowermost stratosphere, but it is too close to our source domain 370-380 K and may not represent the horizontal transport properly. The brief explanation is included in the manuscript. The grey dashed line is added in Fig.5.

(9) *Figure 7: Have the results in this figure been aggregated over the 2010-2013 period? How was the particular interval shown (November-May) chosen? It would be good to define what is meant by*

*"young" in the figure caption as well as the main text. As stated in L239-240, the tropical pathway is more common for tracers released at 350-360 K, but it does not appear to be entirely absent for the 370-380 K tracers in Figure 7c. There are hints of a "fork" in the ASM tracer distribution between ~3.7-4.0 ppmv and ~2% (in which case the cyan arrow may be slightly misplaced). There may even be a faint hint of similar structure for the NASM tracer (Figure 7d), but the cyan arrow, useful though it is, obscures it.*

A. The results in Fig.7 were based on 2010-2011. We didn't correlate the tracer and water vapor from the whole period because the annual variability makes the structure very obscure. We do check the results year by year, they all show similar structures. The text in the manuscript is changed, and a brief explanation is included. We choose the time period November-May considering the transit time (see also Ploeger et al., 2017 and Vogel et al., 2019) from monsoon regions to the tropical pipe. The definition of "young" air mass is included in the caption as well. We agree the tropical pathway is not totally absent for 370-380 K. Although the air in 370-380 K layer over monsoon regions is less leaky than the air in 350-360 K, there is still the possibility of horizontal transport through tropics. This point is added in the revised version of the manuscript.

(10) *L262-263: Is this time difference consistent with known upwelling rates? (A reference would be good.)*

A. The upwelling rate is about 10-20 days/10 K. The references are added in the draft.

(11) *Figure 8: I understand that scaling the standard deviations improves the legibility of the plot, but multiplying by 0.2 seems like a fairly drastic step that produces a misleading impression of the degree of variability. How can such a substantial reduction in the scatter in this plot be justified? If the full envelopes were presented, results for the various destination regions would likely overlap significantly. As it is, I fear that the figure instills more confidence in the separability of the regions than is really warranted.*

A. We agree the scale of 0.2 is too small. Indeed, the results are substantially overlapped if we plot the original standard deviation. It is difficult to see the structures. In the revised manuscript, we multiply the standard deviation by 0.5. We can see the overlap results and the mean time series clearly.

(12) *L314-318: To my eye, the TE into the LS-SH is never dominated by ASM or NASM sources for tracers released at 370-380 K − after February, the curves for all three sources lay nearly on top of one another. Moreover, for the ASM tracers transport from the 350-360 K layer dominates over that from the Tropics starting in December, not January. Finally, the TE from the ASM is nearly 50% larger than that from the NASM, so perhaps "slightly" should be deleted in L318.*

A. We made a mistake to calculate the air mass of the tropical source domain, it affected the calculation of the transport efficiency from tropics, especially for the tropical source tracer released at 370-380 K. We combined the comments from the reviewer and rephrased the paragraph. "The ASM tracers transport from the 350-360 K layer dominates over that from the Tropics starting in December." is changed in the text. The result related to "slightly" is removed.

(13) *L340-341: It would be appropriate to include here some references for the effects of Rossby wave breaking and eddy shedding on mixing monsoon air into the extratropics.*

A. Several references regarding this point are added.

(14) **_L404-411: I was confused the first couple of times that I read this paragraph, because I expected the results cited here to have been shown in Figure 11 - it is the last figure in the paper and freshest in readers' minds when they arrive at the Conclusions. I hadn't understood what was meant by "ultimate" in L404 (in fact, I don't think that the usage of that word conveys quite what the authors intend), and so it took me several minutes to realize that the numbers being quoted here for the most part refer to the end of the simulation period in Figures 8 or 9 and thus do not match the values in Figure 11. I concede that I obviously was not reading these sentences carefully enough, but I'm guessing that many readers may do the same and also may fail to note that Figure 11 shows the "maximum" contributions/efficiencies. That information is noted in the figure caption, but it is not stated when this figure is introduced in L358, which instead describes it as showing "overall contributions, efficiencies, and transit times". In addition, stating values such as 0.9 for the TE in L410 without specifying that this value refers to the end of the simulation compounds the confusion, as does stating a range for the TE from the Tropics to the TrP. In my mind this entire discussion needs to be clarified, with a bit more hand-holding to help the reader follow the details. However, this brings up a philosophical question about whether showing the maximum contributions/efficiencies is really the best approach for Figure 11. Moreover, while reading this paragraph I also wondered why a similar panel for the Tropics was not included in that figure._**

A. We explain the meaning of "ultimate" time in the text. We think that the maximum and ultimate contribution and efficiency of transport from the source regions to the destination regions may help to explain the transport of pollutants with different lifetime to the stratosphere. For better comparison, we summarize the results of the transport from three source regions (ASM, NASM, and Tropics) to three destination regions (LS-NH, LS-SH, and TrP) at the end of the draft.

**3. Minor points of clarification, wording suggestions, and grammar / typo corrections:**

(1) **_L30: influences –> influence_**

A. The text is changed.

(2) **_Figure 1 caption: I questioned the need for the seemingly unimportant detail about the map being produced by python in my initial access review, and I still don't see why this information is useful to the reader. A similar comment applies to Figure. 10._**

A. The sentence about the map information in Fig.1 and Fig.10 is removed.

(3) **_L57: "Meanwhile" seems like an odd choice of word here_**

A. The sentence is changed.

(4) **_L93: add a comma after "anticyclone"_**

A. A comma is included.

(5) **_L116: TrP has already been defined (L41)_**

A. We write the full term Tropical pipe (TrP) together with the extratropical lowermost stratosphere in the Southern Hemisphere (LS-SH) and the extratropical lowermost stratosphere in the Northern Hemisphere (LS-NH) to make the definition of the destination regions more clear. To avoid repetitive definition, we remove the first TrP in the introduction.

(6) *Figure 3 and caption. Although it is stated in the main text, it would be good to add "in July and August" somewhere in the caption, perhaps after "initialized" or before "in CLaMS-E1". Also, some odd glitches are apparent in the dashed line in this figure, especially in panel 3b at about (45N, 10m).*

A. The simulation period " in July and August" is added in the caption of Fig.3, the figures are also changed.

(7) *L155: The interhemispheric difference is fairly small, especially for the total column, so I suggest adding "slightly" in front of "larger"*

A. "slightly" is added in front of "larger".

(8) *L156: since this sentence is about the SH, just to be really clear, add "boreal" in front of "monsoon"*

A. "boreal" is added in front of "monsoon".

(9) *L158-159: portion . . . enters . . . and is (not "enter" and "are")*

A. The text is changed.

(10) *L173: add a comma after "simulations"*

A. A comma is added after "simulations".

(11) *L177: "not shown" – is this point not shown by comparison of Figures 3 and 4?*

A. We can see this point by the comparison of Fig.3 and Fig.4. We write "not shown" because we did not include the figure about the inter-hemisphere difference directly. We remove "not shown" in the revised version of the draft.

(12) *Figure 5 caption: I think it would be helpful to add "over the July 2010 to April 2014 period" after "sections".*

A. "during July 2010 to June 2014" is added in the caption of Fig.5.

(13) *L210:. . . tracers is slightly lower –> . . . tracers is slightly weaker*

A. The text is changed.

(14) *L221: "spread out" might be better than "widespread"*

A. Thank you for the suggestion. We choose "widespread" to emphasize the wider transport in CLaMS-M2 compared to CLaMS-EI.

(15) *L239: ASM (NASM) –> ASM (NASM) region*

A. "region" is added.

(16) *L255: show –> shows*

A. The text is changed.

(17) *L264-266: it would draw the contrast (and flow) better to move "after three months" to right after "However," at the beginning of the sentence.*

A. The sentence is rephrased.

(18) *Figure 8 caption: I think it might work better to say ". . . simulations of air mass fractions (in %) in three source regions"*

A. The caption is changed as suggested.

(19) *L268: that –> those*

A. The text is changed.

(20) *L271: it might be good to add "throughout the year" at the end of this sentence*

A. "throughout the year" is added at the end of the sentence.

(21) *L279-280: it might be good to add "As for the ASM," at the beginning of this sentence*

A. "As for the ASM," is added at the beginning of the sentence.

(22) *L289: it might be good to add "Much" in front of "more air"*

A. "Much" is added in front of "more air".

(23) *L291: delete "and"*

A. "and" is deleted.

(24) *L297: delete "up to and"*

A. "up to and" is deleted.

(25) *L321: To me, "after March" means "starting in April", but in fact the NASM TE exceeds the tropical TE in the TrP region at the beginning of March for the 350-360 K tracers. Thus "after March" should be "by March". Similarly, "after April" should be "by April". In addition, there is a typo at the end of this line: 380 KIn –> 380 K. In*

A. The text is changed to "by March". Because of the mistake about calculating the mass of tropical domain, the results were rephrased here. The typo is corrected.

(26) *L325-326: that –> those. Also, the CLaMS-M2 figure is omitted so I cannot judge myself, but I assume that a similar issue to the point raised above exists for "after December . . . or January".*

A. The text is changed. The point about "after December . . . or January" is checked and revised.

(27) *L337-339: these two sentences are somewhat redundant and could be combined for efficiency (and to eliminate the slightly awkward construction ". . . Fig. 10. Figure 10 . . ."). Also, when were the results for 24 August 2012 shown in this figure initialized?*

A. The two sentences are rephrased and combined to one sentence. The results on 24 August 2012 are based on the tracers initialized from 1 July 2012 to 24 August 2012.

(28) *L342-343: replace the second instance of "CLaMS-EI and CLaMS-M2" in this line with "the two simulations"*

A. The text is changed.

(29) *L366: it would be good to remind readers of these pathways by adding "(monsoon and tropical)" after "pathways"*

A. "(monsoon and tropical)" is included in the after "pathways".

(30) *L371: It is very confusing to start this sentence with "As for the NASM". This kind of construction is often used to set up a discussion of similarity, but the previous sentence is also talking about the NASM, so that doesn't make sense. You may have meant "As is the case for the ASM", in which case there is a typo ("NASM" should be "ASM"). That's what I assumed the first time I read this sentence, so I suggested making that change in my access review. Since the phrase remains in this*

*version, I am guessing that was not your intention, and thus it is probably best to simply delete this phrase.*

A. I didn't understand your access review correctly. It is a typo here. We remove this sentence to make the point less confusing in the revised manuscript.

(31) *L375: maybe add "(not shown)" again at the end of the sentence*

A. "(not shown)" is added at the of the sentence.

(32) *L386: I feel that the Conclusions section starts too abruptly – it needs some sort of introductory sentence to set the stage and sum up what was done in the paper. On the other hand, such a sentence is not really needed at the beginning of the Discussion section. Thus I suggest moving the first sentence in that section ("We have investigated. . .", L330-331) here.*

A. The paragraph about the beginning of "Discussion" and "Conclusions" are changed as suggested.

(33) *L389-390: "vertical differences" is awkward. I suggest instead "differences in the dynamical situation with altitude"*

A. The text is changed as suggested.

(34) *References: the doi's for many of the references are repeated.*

A. The "References" are checked and corrected.